# On Reasoning Strength Planning
# in Large Reasoning Models

**Leheng Sheng**[1]  **An Zhang**[2*] **Zijian Wu** [1]  **Weixiang Zhao**[3]  **Changshuo Shen**[2]  **Yi Zhang**[2]
**Xiang Wang**[2]  **Tat-Seng Chua**[1]

[1]National University of Singapore
[2]University of Science and Technology of China
[3]Harbin Institute of Technology

`leheng.sheng@u.nus.edu, anzhang@u.nus.edu, zijianwu0522@gmail.com,`
`wxzhao@ir.hit.edu.cn, stephen_shen@mail.ustc.edu.cn,`
`zy1230@mail.ustc.edu.cn, xiangwang1223@gmail.com, dcscts@nus.edu.sg`

## Abstract

Recent studies empirically reveal that large reasoning models (LRMs) can automatically allocate more reasoning strengths (*i.e.,* the number of reasoning tokens) for harder problems, exhibiting difficulty-awareness for better task performance. While this automatic reasoning strength allocation phenomenon has been widely observed, its underlying mechanism remains largely unexplored. To this end, we provide explanations for this phenomenon from the perspective of model activations. **We find evidence that LRMs pre-plan the reasoning strengths in their activations even before generation, with this reasoning strength causally controlled by the magnitude of a pre-allocated directional vector.** Specifically, we show that the number of reasoning tokens is predictable solely based on the question activations using linear probes, indicating that LRMs estimate the required reasoning strength in advance. We then uncover that LRMs encode this reasoning strength through a pre-allocated directional vector embedded in the activations of the model, where the vector's magnitude modulates the reasoning strength. Subtracting this vector can lead to reduced reasoning token number and performance, while adding this vector can lead to increased reasoning token number and even improved performance. We further reveal that this direction vector consistently yields positive reasoning length prediction, and it modifies the logits of end-of-reasoning token `</think>` to affect the reasoning length. Finally, we demonstrate two potential applications of our findings: overthinking behavior detection and enabling efficient reasoning on simple problems. Our work provides new insights into the internal mechanisms of reasoning in LRMs and offers practical tools for controlling their reasoning behaviors. Our code is available at https://github.com/AlphaLab-USTC/LRM-plans-CoT.

## 1  Introduction

Large reasoning models (LRMs) [1–3] have demonstrated exceptional performance across a variety of complex reasoning tasks, such as mathematical problem solving [4–6], code generation [7, 8], and scientific question answering [9]. Here we take a closer look at LRMs' ability to allocate reasoning strength commonly quantified by the number of reasoning tokens generated during inference. Increasing reasoning strength has been shown to substantially improve model performance on

---

*An Zhang is the corresponding author.

39th Conference on Neural Information Processing Systems (NeurIPS 2025).

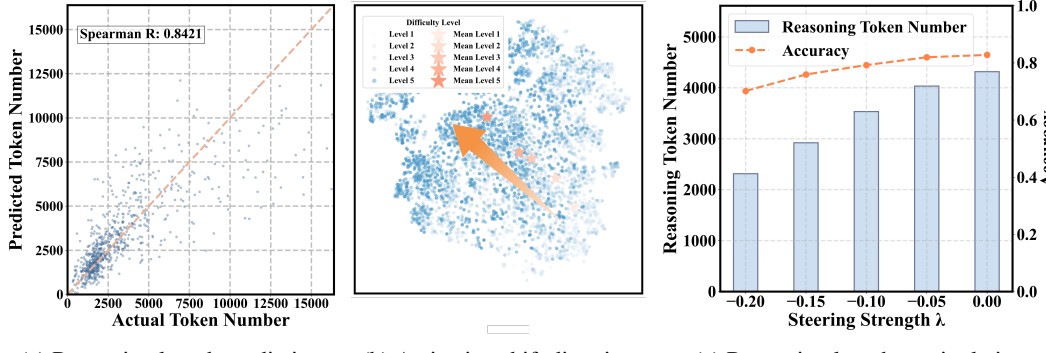

(a) Reasoning length prediction     (b) Activation shift direction     (c) Reasoning length manipulation

Figure 1: (1a) The reasoning length is predictable before the generation of the first reasoning token. (1b) The activations of questions shift towards a pre-allocated direction as difficulty increases. Orange stars denote mean activations of different difficulty levels. (1c) Steering activations of LRMs with this direction vector can causally affect the reasoning token numbers, thereby affecting the performance.

complex reasoning tasks [2, 10]. As a result, it has emerged as a critical factor for both performance optimization and controllable model behavior [11–13].

Recent findings have revealed two key properties of reasoning strength in LRMs: it can be automatically allocated based on problem difficulty [1, 14, 2, 15] and it can be manually controlled through intervention [16–20]. On the one hand, LRMs tend to allocate more reasoning tokens to harder questions, reflecting an implicit adaptation to task complexity [12, 21–24]. On the other hand, reasoning strength can also be explicitly manipulated by prompting a desired length, typically enabled through supervised fine-tuning (SFT) [17] or reinforcement learning (RL) [16] with strength-aware objectives. These empirical observations suggest that LRMs may have automatic and controllable mechanisms for reasoning strength modulation. However, the underlying nature and internal structure of these mechanisms remain largely unexplored.

To fill this research gap, we investigate the underlying mechanism of reasoning strength in LRMs by asking two key questions: (1) Do LRMs pre-plan their reasoning strength before generation? (2) If so, in what form is this control encoded in advance? We approach these questions from the perspective of model activations — that is, how the activations (*i.e.,* latent representations of the question prompt) vary in response to different levels of reasoning strength. For the first, we examine whether the number of reasoning tokens can be predicted solely from the activations corresponding to the input question. For the second, we explore how LRMs encode pre-allocation signals within activations that modulate reasoning strength. Specifically, we employ a linear probe to predict the reasoning token count from question activations, and further extract a pre-allocated direction vector (*i.e.,* pre-allocation vector), such that manipulating this vector within the activation space enables control over the reasoning strength. Positive findings would indicate that LRMs plan reasoning strength ahead of generation through specific pre-allocated activations.

Here we conduct preliminary experiments using DeepSeek-R1-distilled-Qwen on the MATH [5] dataset, which contains math questions across five difficulty levels. As visualized in Figure 1, we summarize three key empirical findings from the activation distribution of the LRM:

- Figure 1a shows that a well-trained linear predictor can estimate the number of reasoning tokens from input activations, achieving a correlation of 0.84 between predicted and actual values. This result indicates that the number of reasoning tokens is predictable prior to generation, suggesting that LRMs pre-plan their reasoning strength in advance.

- As the question difficulty increases, activations consistently shift in a shared direction, as Figure 1b depicts. The activations of math problems exhibit a consistent trend shifting towards the same direction. Specifically, mean difference vectors between high- and low-difficulty questions consistently point in a similar direction, with magnitudes that correlate with question difficulty.

- As shown in Figure 1c, manipulating activations along this direction vector — with varying magnitudes — causally modulates reasoning strength, leading to corresponding changes in LRM performance on reasoning tasks.

These findings suggest that LRMs pre-plan their reasoning strength through pre-allocating a direction vector, whose magnitude encodes the intended strength. To further investigate, we show that this direction vector consistently produces positive predictions of reasoning token numbers, closely aligning with actual values. Moreover, we observe that manipulating activations along this direction influences the logit of the end-of-reasoning token `</think>`, indicating a causal role in terminating the reasoning process. Based on these findings, we further discover two possible potentials of such underlying mechanism of reasoning strength: overthink detection with the predictor and efficient reasoning with activation steering [25].

## 2 Related Works

### 2.1 Large Reasoning Models

Large reasoning models (LRMs) [2, 1] have recently emerged as a new paradigm of large language models for complex task solving through step-by-step reasoning [26, 27, 22]. These models conduct explicit reasoning processes between special tokens of `<think>` and `</think>` before producing final answers [10]. Recent studies have shown that LRMs can adaptively allocate reasoning strength (*i.e.,* the number of reasoning tokens) based on problem difficulty—they tend to allocate more reasoning strength to harder questions to improve accuracy [12, 21–23]. Moreover, it is even possible to specify the reasoning strength via prompting a desired number of reasoning tokens, which can be realized through post-training with length-aware objectives [16, 17]. These phenomena suggest that LRMs may possess some underlying mechanisms to plan and control the strength of their reasoning.

### 2.2 Planning in Language Models

Recent works show that, despite being trained solely on next-token prediction [28], large language models (LLMs) exhibit certain planning capabilities [29–32]. For example, there is evidence that LLMs can anticipate future tokens—such as planning rhyme schemes several lines ahead when composing poems [31]. Moreover, studies have found that models may even pre-plan answer confidence levels or the choices in multiple-choice questions [32]. Despite these findings, the underlying mechanisms behind such planning capabilities remain largely unexplored, and it is still unclear whether similar planning capabilities occur in LRMs. Inspired by these observations, this work presents the first investigation into the reasoning planning capabilities of LRMs, and uncovers the underlying mechanisms that control such planning.

### 2.3 Activation Steering with Linear Direction

Activation steering is one research line among the representation learning [33–39]. Recent advances in mechanism explainability reveal that there exist linear directions inside the activation space of language models that control specific semantical behaviors [40–42]. These directions are typically derived from activation differences between contrasting semantics, such as refusal versus compliance in responses [42]. Manipulating directions via activation addition or subtraction enables behavior modification of language models during inference. It has been evidenced that response style [25, 43, 44], refusal behaviors [42, 45, 46], and memory extraction capabilities [47] have been encoded in linear directions. Recent studies find evidence that by concatenating question prompts with their corresponding chain-of-thought (CoT) answers and computing activation differences between responses of varying reasoning token numbers, it is possible to identify linear directions that control the length of reasoning [19, 48, 49]. In this work, we take an important step further, by demonstrating that such directions have been pre-allocated by LRMs upon observing the question for reasoning strength control, even before generating the answer.

## 3 Reasoning Strength in LRMs is Pre-Planned

In this section, we explore the hypothesis that LRM plans the reasoning strength (*e.g.,* the length of the reasoning process) even before the beginning of the reasoning process [41, 32]. We use linear probing to test this hypothesis [50, 41, 51, 32], predicting the length of reasoning with the activations of the question solely. A good probing result will support our hypothesis [50].

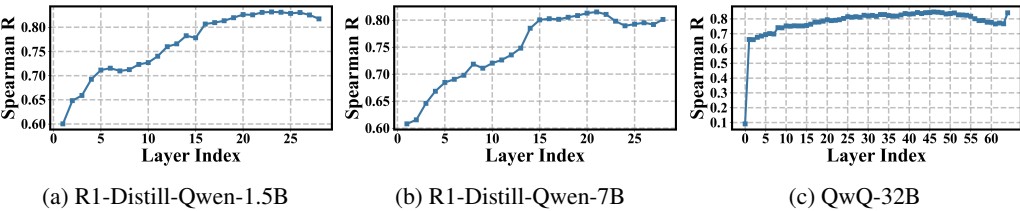

| (a) R1-Distill-Qwen-1.5B | (b) R1-Distill-Qwen-7B | (c) QwQ-32B |

Figure 2: Layer-wise linear regression results

## 3.1 Experimental Setup

**Linear probing.** For each question, we extract the $d$-dimensional residual stream activation $\mathbf{h}^{(l)} \in \mathbb{R}^d$ at the position of the start-of-reasoning `<think>` token, and aim to predict the subsequent reasoning token number $\mathbf{y} \in \mathbb{R}$ using a linear regression model based on $\mathbf{h}^{(l)}$ at each layer $l$. We calculate $\mathbf{y}$ as the number of tokens between the start-of-reasoning token `<think>` and the end-of-reasoning token `</think>`. Formally, given a dataset of $n$ samples, we construct an activation matrix $\mathbf{H}^{(l)} \in \mathbb{R}^{n \times d}$ where each row corresponds to the activation of one question, and corresponding scalar reasoning token number $\mathbf{Y} \in \mathbb{R}^n$. We then learn a linear regression function $\hat{\mathbf{Y}} = \mathbf{H}^{(l)}\mathbf{W}^{(l)} + \mathbf{b}^{(l)}$ by minimizing the following regularized loss:

$$\hat{\mathbf{W}}^{(l)}, \hat{\mathbf{b}}^{(l)} = \arg \min_{\mathbf{W}^{(l)}, \mathbf{b}^{(l)}} \left\| \mathbf{Y} - (\mathbf{H}^{(l)}\mathbf{W}^{(l)} + \mathbf{b}^{(l)}) \right\|_2^2 + \alpha \left\| \mathbf{W}^{(l)} \right\|_1. \tag{1}$$

Equation (3) denotes the Lasso regression [52], where $\mathbf{W}^{(l)} \in \mathbb{R}^d$ and $\mathbf{b}^{(l)} \in \mathbb{R}$ are the learnable parameters of this linear regression. A regularization term $\lambda \left\| \mathbf{W}^{(l)} \right\|_1$ is introduced for avoiding overfitting, where $\alpha$ is a hyperparameter controlling the regularization strength. To implement this Lasso regression, we use the Python package of `scikit-learn` [53]. The illustration and details of this probing process can be found in Appendix B.

**Datasets.** We conduct the linear regression experiments on the MATH [5] dataset, where math questions are divided into five groups according to their difficulty. We randomly split the dataset with a ratio of 9:1 for training and testing.

**Models.** We conduct experiments on a wide range of open-source LRMs, including the distilled R1 model series [2] and the QwQ model [3]. The models we evaluate span a variety of scales, ranging from 1.5B to 32B parameter sizes.

## 3.2 Results

Based on the linear regression experiments, we have the following observations:

**LRMs plan their reasoning strength even before the generation of the first reasoning token, and this planning capability becomes more evident as the layer depth increases.** We visualize the layer-wise prediction results in Figure 2. As shown in this figure, our linear probe can yield high prediction results with correlation coefficients over 0.8 across a range of model sizes and different model kinds, suggesting that the reasoning strength planning is possibly encoded in the model's internal activations before the generation of the first reasoning token. Moreover, as the layer becomes deeper, the prediction results become better. This indicates that the reasoning planning capabilities may be developed in the later layers of these models. The above observations suggest that LRMs may have the capability of planning the reasoning strength in advance. We provide more similar experimental results in the Appendix B.

## 4 LRMs Encode Reasoning Strength via Pre-allocated Direction Vectors

We investigate the underlying mechanism behind this planning capability, given the observation that LRMs may plan their reasoning strength in advance as revealed above. Specifically, inspired by the emerging phenomenon that linear representations can control specific behaviors in language models [40, 42], we hypothesize that LRMs may modulate their reasoning planning through pre-allocated direction vectors embedded within their activation space. We organize this section as follows: We

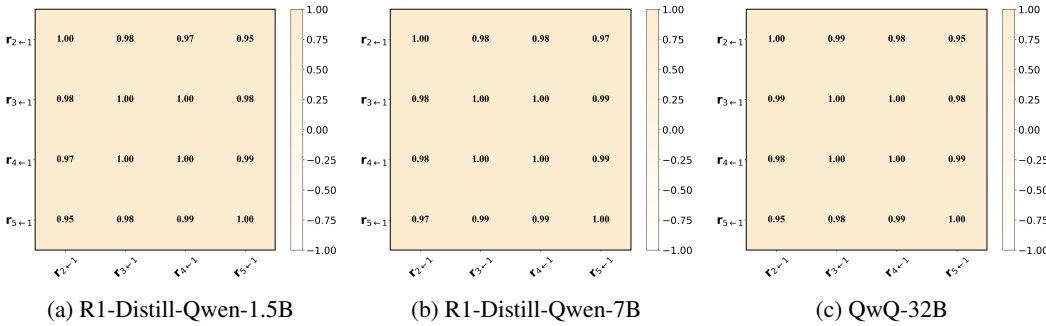

(a) R1-Distill-Qwen-1.5B          (b) R1-Distill-Qwen-7B          (c) QwQ-32B

Figure 3: Cosine similarity between pre-allocated vectors across different difficulties. These vectors exhibit extremely high cosine similarities, indicating LRMs pre-allocate single direction vector for distinguishing different question difficulties.

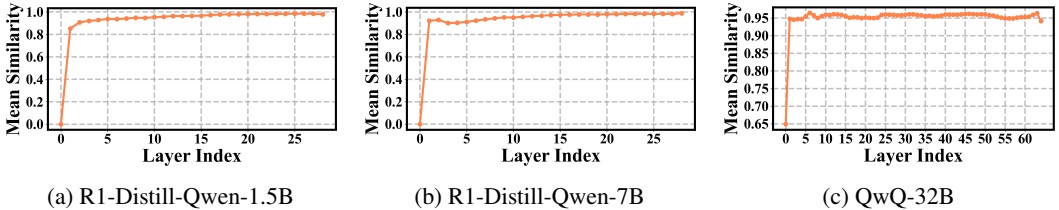

(a) R1-Distill-Qwen-1.5B          (b) R1-Distill-Qwen-7B          (c) QwQ-32B

Figure 4: Layer-wise cosine similarities between four pre-allocated vectors

first describe how to find the existence of such pre-allocated direction vectors using the difference-in-means approach [54] in Section 4.1. After that, in Section 4.2, we reveal that such pre-allocated direction vectors are indeed used for planning reasoning strengths, through their causal effects with activation steering. Then, in Section 4.3, we point out that such pre-allocated direction vectors can be used for predicting the reasoning strengths with the linear predictors we obtained above. Finally, in Section 4.4, we uncover that the mechanism of length planning is ultimately achieved by adjusting the logits of the end-of-think token `</think>` with such pre-allocated direction vectors.

## 4.1 Pre-allocated Direction Vectors Exist for Reasoning Strength Planning

In this section, we test the existence of pre-allocated direction vectors (*i.e.,* pre-allocation vectors) for reasoning strength control. Motivated by the observation that LRMs automatically allocate longer reasoning strength for more difficult questions, we suspect that LRMs use linear activation directions for this control. Therefore, we find such vectors using the difference-in-means approach between questions of varying difficulties.

**Difference in Means [54].** The difference-in-means method [54] effectively extracts activation direction vectors associated with specific model behaviors—such as refusal to answer [42]—by computing the difference between the mean activations associated by two contrasting behaviors of input data pairs (*e.g.,* refusal and compliance). In our case, we construct contrasting data pairs with questions of different difficulties, since LRMs behave in automatically allocating more reasoning strengths on harder tasks. In this way, we may isolate direction vectors related specifically to reasoning strength control, by applying the difference-in-means method. Specifically, we compute the difference-in-means vector $\mathbf{r}_{i \leftarrow 1}^{(l)}$ between difficulty the hardest level $i$ and easiest level 1 on the MATH dataset [5] as:

$$\mathbf{r}_{i \leftarrow 1}^{(l)} = \frac{1}{|\mathcal{D}_i|} \sum_{\mathbf{h}^{(l)} \in \mathcal{D}_i} \mathbf{h}^{(l)} - \frac{1}{|\mathcal{D}_1|} \sum_{\mathbf{h}^{(l)} \in \mathcal{D}_1} \mathbf{h}^{(l)}, \tag{2}$$

where the first and second terms denote the mean activations at layer $l$ computed over the activation sets $\mathcal{D}_i$ and $\mathcal{D}_0$, which correspond to math questions of difficulty level $i$ and 0, respectively. Here, these activations are also extracted at the start-of-reasoning token `<think>` position before generation. By varying the target difficulty $i$ from 1 to 5, we can get four such vectors at each layer, namely $\mathbf{r}_{5 \leftarrow 1}^{(l)}$, $\mathbf{r}_{4 \leftarrow 1}^{(l)}$, $\mathbf{r}_{3 \leftarrow 1}^{(l)}$, and $\mathbf{r}_{2 \leftarrow 1}^{(l)}$. These vectors capture the activation shift from a baseline level of difficulty

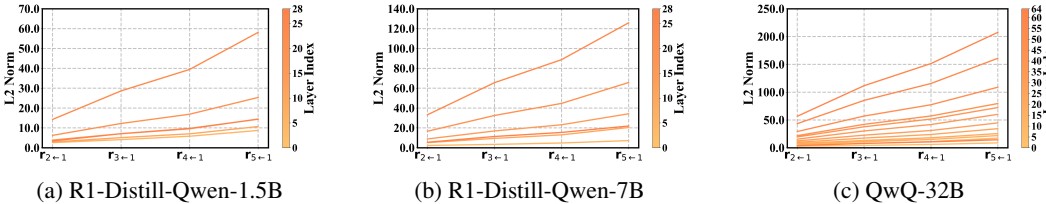

Figure 5: L2 norms of four pre-allocated vectors. The norm becomes bigger as the difficulty increases.

1 to increasingly harder questions. These vectors are pre-allocated since we extract them before generation. If the LRM plans the reasoning strength via a shared directional vector, then the four vectors are expected to show a high degree of similarity.

By analyzing these vectors, we have the following findings:

- **Pre-allocated direction vectors exist for distinguishing questions across different difficulties, since all constructed vectors exhibit consistently high cosine similarities across layers.** We visualize the pairwise cosine similarities among the four extracted vectors in Figure 3, which are taken from the layer with the highest averaged similarity. As shown in these figures, the vectors exhibit extremely high directional consistency, with cosine similarities around 0.99. This suggests that LRMs may utilize a single, shared directional vector to distinguish between questions of different difficulty levels. In addition, we present the trend of average cosine similarity across layers in Figure 4. The results show consistently high similarity scores (*i.e.,* above 0.9), which further increase with layer depth and approach near 1.0 in the final layers.

- **The magnitudes of these pre-allocated vectors highly correlate with the required reasoning token number, showing implicit connections.** Given that the extracted four vectors exhibit nearly identical directions, their magnitudes (*i.e.,* the L2 norm) become the key factor in distinguishing questions of varying difficulty. We plot the magnitudes of these four vectors across different layers in Figure 5. As shown, the vector magnitude increases with question difficulty, and is approximately proportional to the average additional reasoning token number required to solve the question (See more details in Appendix C). This suggests a strong positive correlation between the pre-allocated vector magnitudes and the reasoning strengths allocated by the model.

### 4.2 Pre-allocation Vectors Causally Affect the Reasoning Strengths

To further examine whether these direction vectors are used for reasoning strength planning, we test their causal effect on reasoning strengths via intervention of activation steering [25]. The key idea of activation steering is to inject direction vectors on the activation of language models, to test whether such direction vectors can causally affect the model behaviors [25, 42]. We conduct such activation steering experiments with the average vector $\mathbf{r}^{(l)}$ of our extracted four vectors (*i.e.,* $\mathbf{r}^{(l)} = \frac{1}{4} \sum_{i=2}^{5} \mathbf{r}^{(l)}_{i \leftarrow 1}$) as:

$$\mathbf{h}^{(l)'} \leftarrow \mathbf{h}^{(l)} + \lambda \mathbf{r}^{(l)}, \tag{3}$$

where $\mathbf{h}^{(l)}$ and $\mathbf{h}^{(l)'}$ are the original and post-steered activations at layer $l$, and $\lambda$ is a hyperparameter controlling the strength of steering. More implementation details can be found in Appendix A. By varying the steering strength $\lambda$, we have following observations (See more results in Appendix C.2):

- **Such pre-allocated vectors are indeed responsible for the reasoning strength planning, since steering with extracted vectors will causally affect the reasoning token number.** We visualize in Figure 6 the change in model response length under different steering strengths from -0.2 to 0.2, with an interval of 0.05. As shown, increasing the negative steering strength progressively decreases the model's reasoning token numbers, while the length of the final answer (i.e., the number of tokens after the </think> token) remains unaffected. This indicates that the pre-allocated vector causally controls the planning of the reasoning token number, rather than the answer token number.

- **Controlling reasoning strengths with this pre-allocation vector causally affects the performance [27, 10, 16].** As shown in Figure 6, reducing the model's reasoning token number generally leads to a drop in performance. This demonstrates that steering with the pre-allocated vector enables a simple yet effective test-time scaling mechanism. Furthermore, applying a positive steering

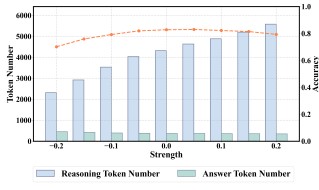
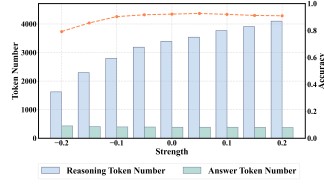
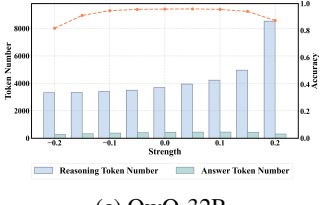

(a) R1-Distill-Qwen-1.5B      (b) R1-Distill-Qwen-7B      (c) QwQ-32B

Figure 6: The causal effect on the reasoning token number and corresponding performance under different steering strength $\lambda$. Decreasing the steering strength $\lambda$ consistently reduces the reasoning token number and corresponding performance.

Table 1: Accuracy % comparison before and after steering across datasets. The performance is bold if improved after steering, and the absolute improvement is shown as superscripts.

|  | MATH500 | AIME2024 | OlympiadBench | Average |
|---|---|---|---|---|
| R1-Distill-Qwen-1.5B | 82.77 | 28.33 | 44.41 | 51.84 |
| + Steering | $\mathbf{83.00}^{+0.23}$ | $\mathbf{29.58}^{+1.25}$ | $\mathbf{44.67}^{+0.26}$ | $\mathbf{52.42}^{+0.58}$ |
| R1-Distill-Qwen-7B | 92.17 | 50.42 | 58.13 | 66.91 |
| + Steering | $\mathbf{92.65}^{+0.48}$ | $\mathbf{55.00}^{+4.58}$ | $\mathbf{58.80}^{+0.67}$ | $\mathbf{68.82}^{+1.91}$ |
| R1-Distill-Qwen-14B | 93.67 | 61.25 | 62.19 | 72.37 |
| + Steering | $\mathbf{94.05}^{+0.38}$ | $\mathbf{67.08}^{+5.83}$ | $\mathbf{63.02}^{+0.83}$ | $\mathbf{74.72}^{+2.35}$ |
| R1-Distill-Qwen-32B | 94.05 | 63.75 | 63.46 | 73.75 |
| + Steering | $\mathbf{94.67}^{+0.62}$ | $\mathbf{67.50}^{+3.75}$ | $\mathbf{64.11}^{+0.65}$ | $\mathbf{75.43}^{+1.68}$ |
| QwQ-32B | 95.90 | 65.42 | 31.25 | 64.19 |
| + Steering | $\mathbf{96.03}^{+0.13}$ | $\mathbf{66.67}^{+1.25}$ | $31.25^{+0.00}$ | $\mathbf{64.65}^{+0.46}$ |

strength can even improve model performance. As shown in Table 1, moderate positive steering shows potential in enhancing performance across multiple math datasets, including MATH500 [5], AIME [55], and OlympiadBench [56]. However, increasing the steering strength beyond a certain point does not lead to further gains, and may even degrade performance. We attribute this to the possible intelligence upper bound of such LRMs. More results are in the Appendix C.2.

## 4.3 Pre-allocation Vectors Yield Positive Reasoning Token Number Prediction

In this section, we discuss the connection between the reasoning token number predictor and these pre-allocated vectors we obtained. We reveal that these vectors tend to generate positive reasoning token number predictions, further proving their role in the reasoning strength control. Specifically, we can estimate the effect of these vectors with the linear predictor we obtained in Section 3 as follows:

$$\hat{\mathbf{y}}^{(l)} = \mathbf{r}^{(l)}\hat{\mathbf{W}}^{(l)} + \hat{\mathbf{b}}^{(l)}. \tag{4}$$

**The pre-allocation vectors yield positive reasoning token number predictions in most cases.** We visualize the predicted reasoning token number across layers when applying the steering vector with a multiplier of 0.2 in Figure 7. The average prediction (*i.e.*, $\hat{\mathbf{y}}^{(l)}$) shows a consistent positive trend, indicating that the steering vector reliably adjusts the reasoning length. Moreover, $\hat{\mathbf{y}}^{(l)}$ aligns well with the actual causal changes shown in Figure 6, highlighting a strong link between regression predictions and the steering-induced effects, thus reinforcing the vector's role in reasoning strength planning. More similar results are in Appendix C.3.

## 4.4 Pre-allocation Vectors Control Reasoning Strengths by Modifying Logits of `</think>`

To study how such pre-allocated vectors affect the reasoning strength, we take one possible perspective: the impact on the logits of the end-of-reasoning token `</think>`. We have the following findings:

**These pre-allocation vectors control the reasoning strength by modifying the logits of the end-of-reasoning token `</think>`.** We visualize the distribution of logits for the `</think>` token under different steering strengths in Figure 8. As shown, applying a negative steering strength results in an overall increase in the logits of the `</think>` token, indicating a higher likelihood of its occurrence, which leads to fewer reasoning tokens. Conversely, positive steering strength decreases the logits of the `</think>` token, reducing the likelihood of generating this token, which leads to more reasoning

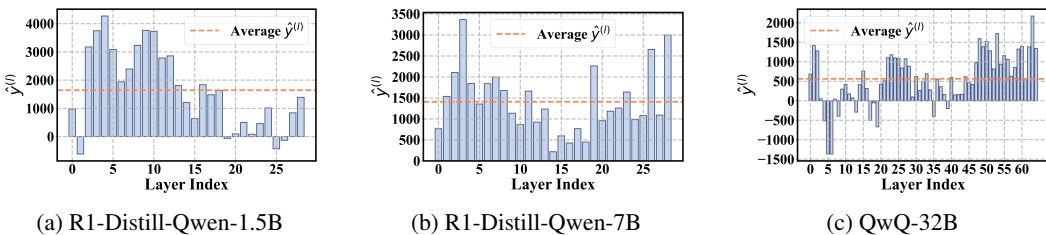

(a) R1-Distill-Qwen-1.5B  (b) R1-Distill-Qwen-7B  (c) QwQ-32B

Figure 7: The predicted reasoning number $\hat{\mathbf{y}}^{(l)}$ yielded by the pre-allocation vector $\mathbf{r}^{(l)}$. Pre-allocation vector yields positive predictions in most cases.

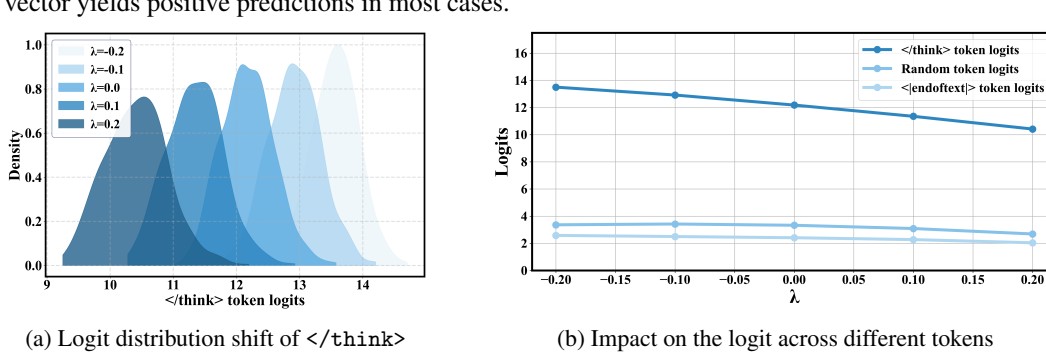

(a) Logit distribution shift of `</think>`  (b) Impact on the logit across different tokens

Figure 8: The effect on the end thinking token `</think>` when steering with different strengths on R1-Distill-Qwen-1.5B. (8a) The pre-allocated vectors control the reasoning strength by causally affecting the logits of end-of-reasoning token `</think>`. (8a) The impact of logits on `</think>` is significantly higher than other tokens.

token numbers. We also compare the impact on the logits with the eos token `<|endoftext|>` and randomly selected tokens. As shown in Figure 8b, the impact on the end-of-reasoning token is significantly higher than random tokens and the eos token `|endoftext|`. These observations suggest that the pre-allocated vectors primarily modulate reasoning strength by adjusting the logits of the end-of-reasoning token. More similar results are in Appendix C.4. We also reveal that the steering mechanism also affects reasoning-related token logits within the reasoning process in Appendix C.5

## 5 Potentials of Our Findings

We aim to investigate whether our findings have potential in more diverse domains, despite our current analysis mainly focusing on mathematical problems. Specifically, we discuss two possible generalized potentials of our findings: overthink detection and efficient inference. It is important to highlight that we merely outline the potential application directions, and more efforts are required to make such potential applicable in real-world practice.

### 5.1 Overthink Detection before Model Generation

In this section, we demonstrate how to detect potential overthink behavior before the generation of LRMs. LRMs exhibit risks in overthink unexpectedly, generating unnecessarily long reasoning traces. Such overthink phenomena can cause significant real-world deployment consequences, not only significantly increasing the computational cost for model providers, but also unnecessarily prolonging the waiting time for users [57, 58, 13]. If we can detect overthink in advance (*i.e.,* before generation), we may reduce both computation and latency by, for example, switching to a lighter model for serving [58]. We argue that we can effectively identify the occurrence of possible overthink, by using our trained predictors to estimate the reasoning strength in advance. We test the predictions differences on data pairs of non-overthink questions and overthink questions. These overthink questions are constructed by forcing LRMs to overthink on vanilla questions by adopting overthink attacks [57], where these vanilla questions are sampled from the AlpacaEval dataset. More details can be found in Appendix D.1. We visualize our prediction results in Figure 9. We can find that our predictor yields significantly longer reasoning lengths on overthink questions than on the vanilla

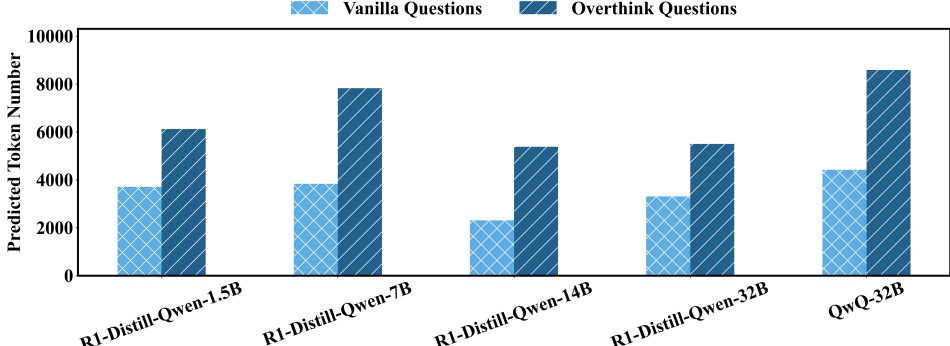

Figure 9: Overthink detection on AlpacaEval [59] dataset. Our predictor can successfully detect the overthink phenomenon by yielding higher predicted reasoning lengths on overthink questions.

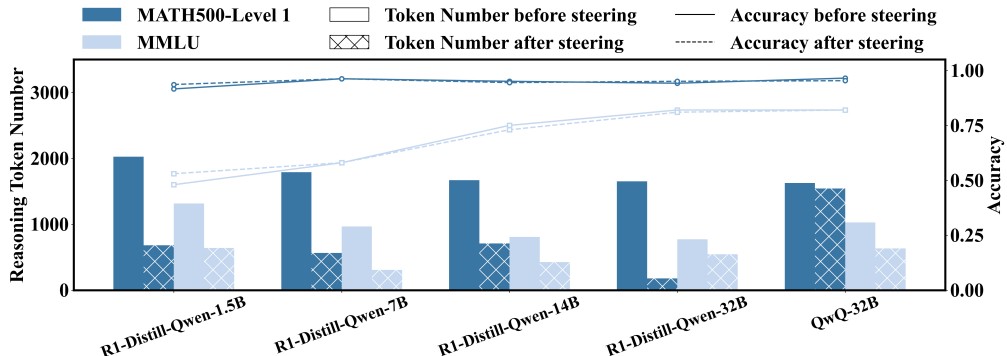

Figure 10: Reasoning token numbers (bar chart) and model performance (line chart on MATH500 [5] and MMLU [60]). The number of reasoning tokens can be significantly reduced without harming the performance of LRMs on easy tasks.

non-overthink questions. **This indicates that we can detect possible overthink phenomena before the generation, which is based on our findings that LRMs plan their reasoning strengths.**

## 5.2 Efficient Inference

In this section, we present how to leverage our findings to achieve more efficient inference, avoiding overthinking on simple questions [13]. To test this potential, we conduct activation steering experiments on two kinds of questions that LRMs tend to overthink: general language understanding dataset MMLU [60] and Level 1 questions on MATH500 [5]. We reduce the number of reasoning tokens by applying activation steering with a proper negative steering strength $\lambda$. We report the performance under such steering experiments in Figure 10. As shown in this figure, we can significantly reduce the number of reasoning tokens on such questions, indicating that LRMs may wrongly allocate reasoning strengths on these easy tasks, which induces their overthinking. While reducing the number of reasoning tokens on these tasks can still maintain the performance of LRMs. This result suggests the potential of using activation steering for efficient reasoning, and also reveals the underlying overthink mechanism of LRMs on easy questions.

## 6  Limitations

This work has several limitations. First, we only use a linear probe and do not explore whether more complex architectures, such as MLPs, could yield better performance in predicting reasoning length. In addition, our experiments focus primarily on the Qwen model series [61], and it remains unexplored whether these findings hold for reasoning models based on other backbones [62].

## 7   Conclusion

This paper investigated whether and how large reasoning models (LRMs) plan reasoning strength (*i.e.,* the number of reasoning tokens) before generating answers. Using linear probing, we showed that the reasoning token number can be predicted merely using the activations of the question, suggesting implicit reasoning strength planning capabilities before generation. We found the existence of pre-allocated direction vectors in LRMs, whose magnitude causally affects the number of reasoning tokens. We further revealed that these vectors may affect the logits of the end-of-thinking token </think> to achieve reasoning strength control. Finally, we discussed two potential applications of our findings: overthink detection before generation and efficient inference. Our paper studied the underlying mechanism of the reasoning strength planning in LRMs from the model activation perspective, which can help the community better understand LRMs [2].

## Acknowledgments

This research/project is supported by the National Research Foundation, Singapore under its National Large Language Models Funding Initiative (AISG Award No: AISG-NMLP-2024-002). Any opinions, findings and conclusions or recommendations expressed in this material are those of the author(s) and do not reflect the views of National Research Foundation, Singapore

---

[2]The broader impacts will be discussed in Appendix F

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

# A  Implementation Details

We implement all the experiments on 8 NVIDIA A100 GPUs. The whole computational resource cost of this research is about 80 A100 GPU days, which is mainly spent on the answer generation.

For the answer generation, we use the vLLM [63] framework for acceleration. Following the suggestions of DeepSeek [3], we set the temperature as 0.6 to prevent endless repetitions, set the maximum new generation length as 16,384, and set the rollout number as 8. We take the average accuracy of all 8 rollouts as the accuracy of one question, and report the final accuracy by taking the average accuracy of each question.

We use the following template for generation:

> **PROMPT:**
> Please reason step by step, and put your final answer within \boxed{}.
> This is the problem:
> {problem}

Figure 11: Prompts used for answer generation.

Here, the {problem} will be replaced by a real question. After we finish generation, we extract the answer inside the \boxed{} for evaluation.

---

[3] https://huggingface.co/deepseek-ai/DeepSeek-R1-Distill-Qwen-7B

# B    Reasoning Strength in LRMs is Pre-Planned

## B.1    Experimental Details

We visualize the linear probing process in Figure 12. We first extract the activation of LRMs $\mathbf{h}^{(l)}$ at the last token position. Then we train a linear regression for predicting the subsequent reasoning token number $\mathbf{y}$, which is calculated through the tokenizer. For reducing the overfitting, we set the regularization term $\alpha$ in Equation 1 as 10.

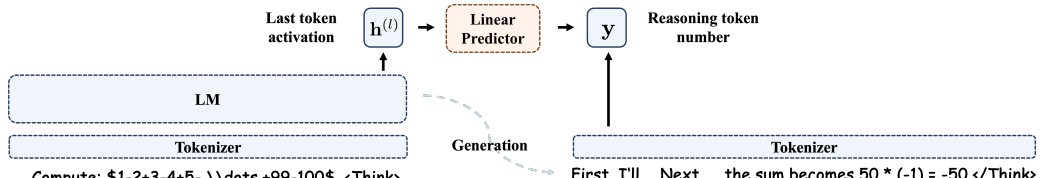

Figure 12: The procedure of linear probing.

## B.2    Layer-wise Regression Results

We visualize the layer-wise regression results on R1-Distill-Qwen-14B and R1-Distill-Qwen-32B in Figure 13. On these two models, the linear regression exhibits the same pattern of an increasing trend as the model depth increases. Similarly, the correlation coefficient reaches over 0.8, indicating that the reasoning strength can be predicted before the model generation.

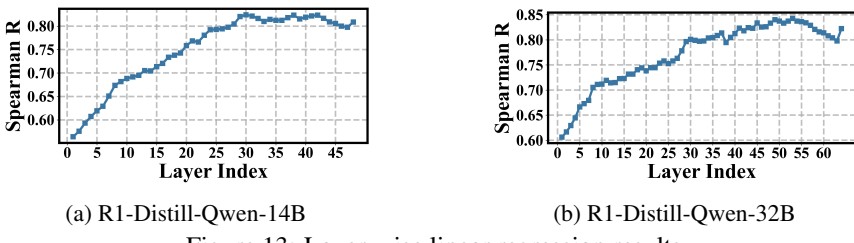

(a) R1-Distill-Qwen-14B          (b) R1-Distill-Qwen-32B

Figure 13: Layer-wise linear regression results

# C   LRMs Encode Reasoning Strength via a Pre-allocated Direction Vector

## C.1   Existence of Pre-allocated Direction Vectors for Reasoning Strength Control

We visualize the cosine similarity matrix of the four extracted vectors from R1-Distill-Qwen-14B and R1-Distill-Qwen-32B in Figure 14. We have a similar observation that all these vectors exhibit extremely high cosine similarities near 1.0. This indicates that LRMs actually use a single direction vector for distinguishing questions of different difficulty levels.

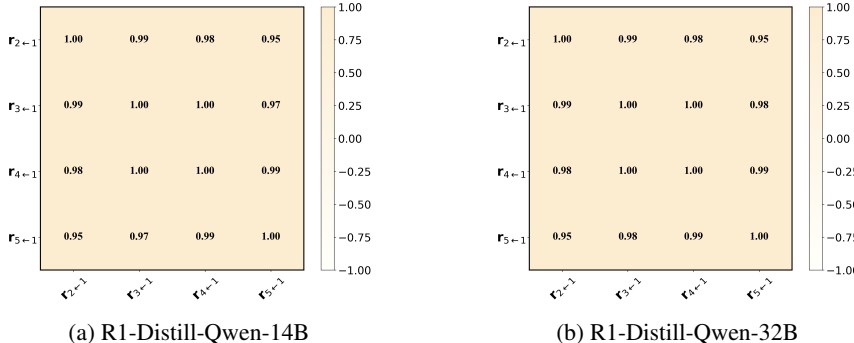

(a) R1-Distill-Qwen-14B                  (b) R1-Distill-Qwen-32B

Figure 14: Cosine similarity between pre-allocated vectors across different difficulties. These vectors exhibit extremely high cosine similarities, indicating LRMs pre-allocate a single direction vector for distinguishing different question difficulties.

We visualize the layer-wise mean cosine similarities between four extracted vectors from R1-Distill-Qwen-14B and R1-Distill-Qwen-32B in Figure 15. We observe that these vectors exhibit consistently high cosine similarities, with an increasing trend as the layer depth increases. Finally, the mean cosine similarity reaches around 1.0, indicating these vectors become a single direction vector in the later layers.

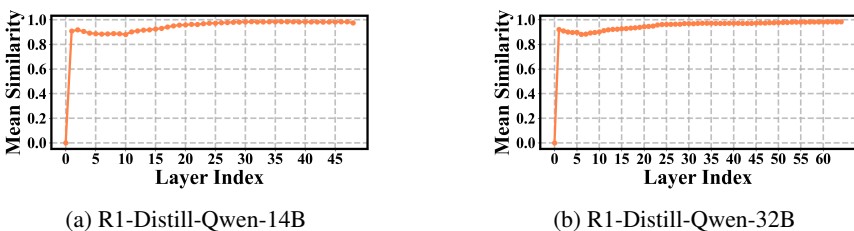

(a) R1-Distill-Qwen-14B                  (b) R1-Distill-Qwen-32B

Figure 15: Layer-wise cosine similarities between four pre-allocated vectors

We visualize the L2 norm of these extracted four vectors from R1-Distill-Qwen-14B and R1-Distill-Qwen-32B in Figure 16. We observe that the norm of these vectors becomes bigger as the difficulty increases. Moreover, this trend is also similar to the increased reasoning token number as the difficulty increases, as shown in Figure 17. This indicates that LRMs use the magnitude of these direction vectors for handling different question difficulties.

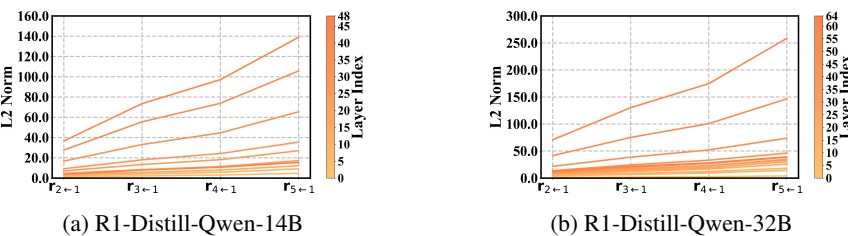

(a) R1-Distill-Qwen-14B                  (b) R1-Distill-Qwen-32B

Figure 16: L2 norms of four pre-allocated vectors. The norm becomes bigger as the difficulty increases.

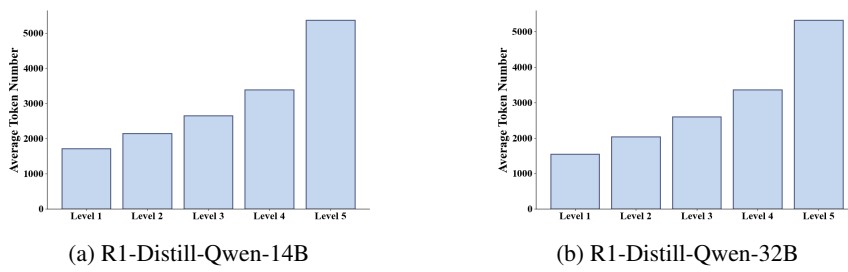

(a) R1-Distill-Qwen-14B        (b) R1-Distill-Qwen-32B

Figure 17: Average reasoning token number for different question difficulties.

## C.2 Pre-allocated Vectors Causally Affect the Reasoning Strengths

We apply the activation steering at each layer and each position of LRMs. We provide more results when steering with the pre-allocation vector $\mathbf{r}^{(l)}$, in Figure 18, Figure 19, and Figure 20. We have similar observations to those in Section 4.2. When steering with negative $\lambda$, we observe a consistent decreasing trend in the reasoning token number and the decreased performance. When steering with positive $\lambda$, we observe a consistent increasing trend in the reasoning token number. However, despite appropriate positive $\lambda$ can improve the performance, this is not consistent as the $\lambda$ increases. We attribute this to the capability upper bound of these LRMs. Moreover, the steering only affects the reasoning token number, while maintaining the answer token number largely unchanged. This indicates that the pre-allocated direction vector is mainly responsible for the reasoning token number.

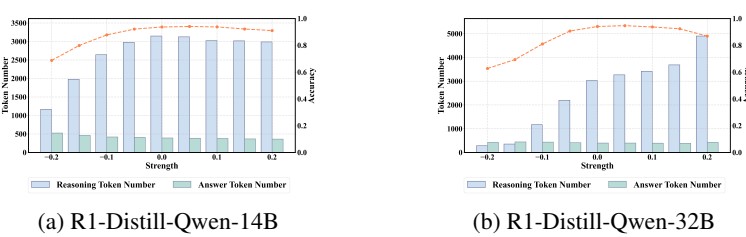

(a) R1-Distill-Qwen-14B        (b) R1-Distill-Qwen-32B

Figure 18: The causal effect on the reasoning token number and corresponding performance under different steering strength $\lambda$ on the dataset MATH500.

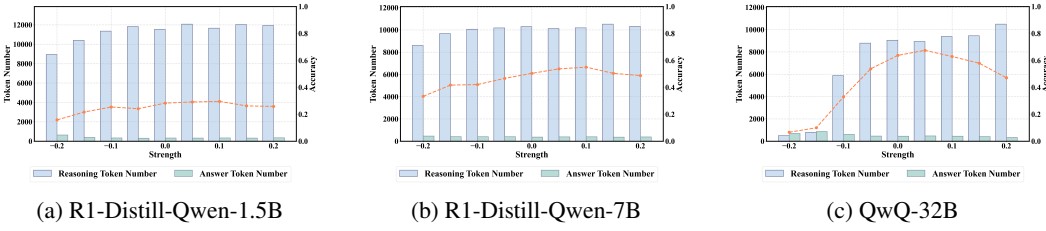

(a) R1-Distill-Qwen-1.5B     (b) R1-Distill-Qwen-7B     (c) QwQ-32B

Figure 19: The causal effect on the reasoning token number and corresponding performance under different steering strength $\lambda$ on the dataset AIME.

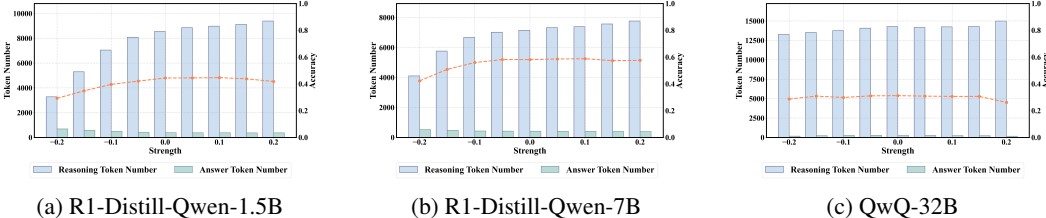

(a) R1-Distill-Qwen-1.5B     (b) R1-Distill-Qwen-7B     (c) QwQ-32B

Figure 20: The causal effect on the reasoning token number and corresponding performance under different steering strength $\lambda$ on the dataset OlympiadBench.

### C.3 Pre-allocation Vectors Yield Positive Reasoning Token Number Prediction

We provide more results in predicting the reasoning token number directly using the pre-allocation vectors in Figure 21. As shown in this figure, in most cases, the pre-allocation vectors yield positive predictions, indicating the close correlation of such vectors with our obtained predictors. This suggests that LRMs are indeed using such pre-allocated vectors for planning their reasoning strength, and the predication also largely relies on these vectors.

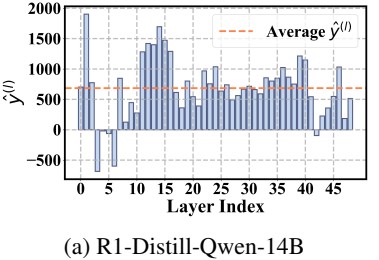 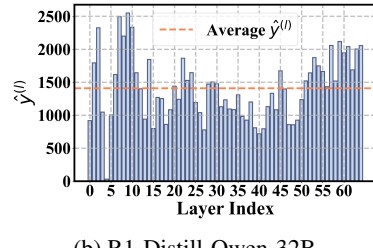

(a) R1-Distill-Qwen-14B      (b) R1-Distill-Qwen-32B

Figure 21: The predicted reasoning number $\hat{\mathbf{y}}^{(l)}$ yielded by the pre-allocation vector $\mathbf{r}^{(l)}$. Pre-allocation vector yields positive predictions in most cases.

### C.4 Pre-allocated Vectors Control Reasoning Strengths by Modifying Logits of `</think>`

We provide more results about how these pre-allocated direction vectors control the reasoning strength by modifying the logits of the end-of-reasoning token `</think>`.

We conduct the same activation steering as we do in Section 4.2, varying the steering strength $\lambda$ from -0.2 to 0.2. Then, we directly extract the logits of each token at the last token position (*i.e.,* the start-of-reasoning `<think>`). We visualize the results in Figure 22. We can observe that, as the steering strength $\lambda$ increases from -0.2 to 0.2, the logits of the end-of-reasoning token `</think>` decrease. This indicates that LRMs are less likely to generate such tokens, thereby leading to more reasoning tokens. Moreover, as shown in Figure 22b, this steering mainly has more impact on the logits of `</think>` than randomly selected tokens and the EOS token `<endoftext>`. Here, the random token logits denote the average logits of 500 randomly selected tokens. This indicates that the steering mainly focuses on adjusting the reasoning strength by manipulating the `</think>`.

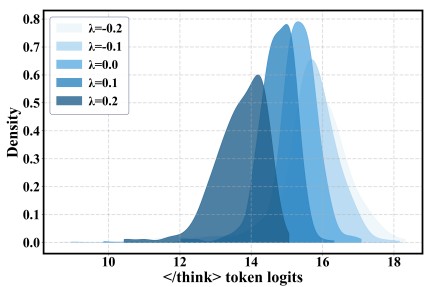 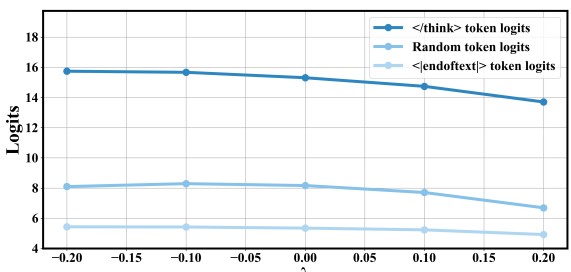

(a) Logit distribution shift of `</think>`      (b) Impact on the logit across different tokens

Figure 22: The effect on the end thinking token `</think>` when steering with different strengths on R1-Distill-Qwen-7B. (8a) The pre-allocated vectors control the reasoning strength by causally affecting the logits of end-of-reasoning token `</think>`. (8a) The impact of logits on `</think>` is significantly higher than other tokens.

### C.5 Pre-allocated Vectors Control Reasoning-related Token Logits

We further test the changes in the logits of reasoning-related tokens. We find that with a positive steering strength, the logits of complex reasoning-related tokens also increase, beyond merely decreased logits of the end-of-reasoning token `</think>`.

We report the changes in the logits for these reasoning-related tokens as follows in Table 2. These tokens are usually regarded as reflection patterns in the R1-style response. Increasing the logits of

these tokens increases the probabilities of reflection and increases of reasoning token numbers, and decreasing them has the opposite effect. As shown in Table 2, positive steering strengths increase the logits of these tokens while negative steering strengths decrease them. This indicates the reasoning strength control is not superficial.

Table 2: Impact of activation steering on logits of reasoning-related tokens.

| Token | $\lambda = -0.2$ | $\lambda = -0.1$ | $\lambda = 0.1$ | $\lambda = 0.2$ |
|---|---|---|---|---|
| Alright | -0.2033 | -0.0638 | +0.0117 | +0.0061 |
| Hmm | -1.1171e-4 | -4.7589e-5 | +2.5699e-5 | +7.8185e-6 |
| Oh | -5.1966e-8 | -2.9514e-8 | +8.5295e-8 | +2.2929e-7 |
| Wait | -7.3789e-9 | -1.6405e-14 | +2.8739e-8 | +9.2974e-8 |

Additionally, to further test whether simply changing the logits of the end-of-reasoning token can achieve the same performance, we conduct experiments by multiplying the logits of the end-of-reasoning token with a factor $\gamma$ (*i.e.*, , $\texttt{logit}_{new} = \gamma * \texttt{logit}_{old}$). We report the results on MATH500 in Table 3. We find that naively changing the logits will lead to unstable manners, either leading to overly increased answer token number (*i.e.*, $\gamma = 2$ on R1-Distill-Qwen-7B) or easily leading to decoding errors (*i.e.*, $\gamma = 0.8$).

These results further suggest that the changes in the logits of the end-of-reasoning token are the result of planning, rather than the cause of planning.

Table 3: Effect of changing the logits with $\gamma$.

| | Reasoning Token Number | Answer Token Number | Accuracy |
|---|---|---|---|
| **R1-Distill-Qwen-1.5B** | | | |
| $\gamma = 2$ | 4445.93 | 376.89 | 82.00 |
| $\gamma = 1$ | 4316.95 | 372.12 | 82.78 |
| $\gamma = 0.8$ | 16159.28 | 2.00 | 0.15 |
| **R1-Distill-Qwen-7B** | | | |
| $\gamma = 2$ | 1013.48 | 5795.13 | 56.20 |
| $\gamma = 1$ | 3392.95 | 386.03 | 92.18 |
| $\gamma = 0.8$ | 11309.18 | 2.00 | 0.00 |
| **R1-Distill-Qwen-14B** | | | |
| $\gamma = 2$ | 3267.30 | 387.06 | 94.35 |
| $\gamma = 1$ | 3148.91 | 391.72 | 93.37 |
| $\gamma = 0.8$ | 15913.95 | 2.00 | 0.15 |
| **R1-Distill-Qwen-32B** | | | |
| $\gamma = 2$ | 3038.15 | 393.72 | 95.10 |
| $\gamma = 1$ | 3029.32 | 395.75 | 94.05 |
| $\gamma = 0.8$ | 6963.46 | 2.00 | 0.00 |
| **QwQ-32B** | | | |
| $\gamma = 2$ | 3623.64 | 423.12 | 95.45 |
| $\gamma = 1$ | 3690.35 | 436.25 | 95.90 |
| $\gamma = 0.8$ | 16256.47 | 1.00 | 0.00 |

# D   Potentials of Our Findings

## D.1   Overthink Detection before Model Generation

In this section, we discuss whether our findings can help us detect the potential overthinking phenomenon even before the generation. To study this, we sample 100 questions from the AlpacaEval [59] dataset as the vanilla questions. We then generate one overthink question for each vanilla question, with the overthink attack [57], which proves to be effective in inducing overthink while maintaining the accuracy on vanilla questions. In this way, we can test whether our predictor can detect the overthink phenomenon in advance, by checking whether the predicted token number on overthink questions is much more than that on vanilla questions. Results in Section 5.1 show that our predictor shows potential in overthink detection.

## D.2   Efficient Inference

### D.2.1   Details

For evaluation efficiency, we sample 100 questions from MMLU [60] and transform them into single-choice questions for LRMs to answer. We adopt the same setting as in lm-evaluation-harness [4] for evaluation.

---

[4] https://github.com/EleutherAI/lm-evaluation-harness

# E  Discussion

## E.1  Generalization on More Domains

We discussed the generalization capabilities of our findings on other general domains beyond MATH, such as AlpacaEval and MMLU in Section 5, where our findings of reasoning strength prediction and strength control also hold on these two datasets. We also add one more experiment on the complex logic reasoning tasks such as GPQA diamond [9] and LiveCodeBench [64] to better demonstrate the generalization capabilities. As shown in Table 4 and Table 5, the number of reasoning tokens generally exhibits a similar pattern with the $\lambda$, indicating the generalization of this pre-allocation vector to other domains. Additionally, proper positive steering can also bring performance improvements. Since we found it hard to extract the generated code in a correct format for evaluation for code generation tasks when varying the response lengths, we just report the effect on the reasoning token numbers. There is only one exception on R1-Distill-Qwen-1.5B, where reducing the reasoning token number even brings the performance gain. We carefully analyze this model and find that when generating overlong responses, it tends to forget to follow the original instruction, and sometimes the results get cut off. Therefore, reducing the reasoning token number helps it avoid such bad issues like cut off, and increasing it will have the opposite effect.

Table 4: The effect of steering and performance on GPQA diamond.

| | | | | | $\lambda$ | | | | |
|---|---|---|---|---|---|---|---|---|---|
| Model | -0.2 | -0.15 | -0.1 | -0.05 | 0 | 0.05 | 0.1 | 0.15 | 0.2 |
| **R1-Distill-Qwen-1.5B** | | | | | | | | | |
| Tokens | 959.44 | 2930.29 | 5659.00 | 6677.72 | 7428.22 | 7690.48 | 7902.20 | 7805.64 | 7784.97 |
| Accuracy | 8.01 | 6.05 | 8.20 | 12.30 | 13.86 | 16.80 | 17.58 | 16.80 | 19.53 |
| **R1-Distill-Qwen-7B** | | | | | | | | | |
| Tokens | 1022.10 | 2755.88 | 5250.92 | 6243.68 | 6998.01 | 7326.04 | 7875.52 | 8103.89 | 8380.32 |
| Accuracy | 5.66 | 12.30 | 22.27 | 28.13 | 33.92 | 36.13 | 40.23 | 39.26 | 40.33 |
| **R1-Distill-Qwen-14B** | | | | | | | | | |
| Tokens | 886.26 | 2289.21 | 3910.04 | 6035.36 | 6365.84 | 6167.72 | 5927.13 | 5463.18 | 5137.19 |
| Accuracy | 15.42 | 17.97 | 23.24 | 44.34 | 50.78 | 51.95 | 51.56 | 51.76 | 50.20 |
| **R1-Distill-Qwen-32B** | | | | | | | | | |
| Tokens | 266.07 | 296.68 | 1612.85 | 4696.14 | 5881.31 | 6391.68 | 6809.91 | 7128.37 | 7274.20 |
| Accuracy | 17.38 | 19.33 | 19.14 | 44.73 | 55.76 | 56.83 | 59.77 | 59.38 | 54.30 |
| **QwQ-32B** | | | | | | | | | |
| Tokens | 6490.33 | 6826.64 | 7368.09 | 7472.53 | 8087.62 | 8811.63 | 8763.01 | 9161.22 | 9862.67 |
| Accuracy | 48.43 | 58.79 | 59.38 | 59.76 | 60.55 | 60.96 | 60.16 | 59.38 | 55.86 |

Table 5: The effect of steering and performance on LiveCodeBench (Code Execution).

| | | | | | $\lambda$ | | | | |
|---|---|---|---|---|---|---|---|---|---|
| Model | -0.2 | -0.15 | -0.1 | -0.05 | 0 | 0.05 | 0.1 | 0.15 | 0.2 |
| **R1-Distill-Qwen-1.5B** | | | | | | | | | |
| Tokens | 1971.90 | 2569.96 | 3112.51 | 3942.85 | 4682.46 | 5856.20 | 7120.98 | 8660.37 | 10482.23 |
| Accuracy | 26.30 | 23.02 | 18.32 | 13.20 | 8.72 | 5.48 | 2.30 | 1.41 | 0.84 |
| **R1-Distill-Qwen-7B** | | | | | | | | | |
| Tokens | 281.09 | 358.46 | 678.20 | 1209.86 | 1625.67 | 2198.33 | 2942.55 | 3701.05 | 4688.05 |
| Accuracy | 61.85 | 66.75 | 72.65 | 79.33 | 79.23 | 79.30 | 74.63 | 80.17 | 80.85 |
| **R1-Distill-Qwen-14B** | | | | | | | | | |
| Tokens | 346.57 | 461.33 | 929.89 | 1267.48 | 1428.35 | 1562.22 | 1745.25 | 1966.76 | 2197.66 |
| Accuracy | 68.42 | 74.01 | 82.05 | 91.54 | 91.03 | 91.28 | 89.35 | 89.35 | 89.67 |
| **R1-Distill-Qwen-32B** | | | | | | | | | |
| Tokens | 230.60 | 345.78 | 501.81 | 827.44 | 1201.67 | 1682.29 | 2317.60 | 2916.74 | 3465.59 |
| Accuracy | 52.14 | 56.89 | 58.51 | 68.58 | 79.54 | 86.69 | 90.03 | 93.63 | 86.01 |
| **QwQ-32B** | | | | | | | | | |
| Tokens | 1053.54 | 1151.99 | 1284.14 | 1439.79 | 1682.31 | 1895.04 | 2202.13 | 2521.79 | 2927.20 |
| Accuracy | 93.74 | 94.99 | 96.45 | 98.38 | 99.06 | 99.27 | 99.27 | 98.64 | 98.49 |

Table 6: The effect of steering and performance on LiveCodeBench (Code Generation).

| Model | -0.2 | -0.15 | -0.1 | -0.05 | 0 | 0.05 | 0.1 | 0.15 | 0.2 |
|---|---|---|---|---|---|---|---|---|---|
| | | | | | $\lambda$ | | | | |
| **R1-Distill-Qwen-1.5B** | | | | | | | | | |
| Tokens | 9176.89 | 9548.69 | 9915.46 | 9999.58 | 10422.80 | 10744.27 | 10963.86 | 11083.30 | 11070.14 |
| **R1-Distill-Qwen-7B** | | | | | | | | | |
| Tokens | 1905.08 | 6409.04 | 8333.01 | 8747.09 | 8964.33 | 9194.85 | 9549.61 | 9635.65 | 10233.36 |
| **R1-Distill-Qwen-14B** | | | | | | | | | |
| Tokens | 6732.97 | 6987.67 | 7578.29 | 7310.26 | 7299.31 | 7238.23 | 7008.38 | 6797.31 | 6494.35 |
| **R1-Distill-Qwen-32B** | | | | | | | | | |
| Tokens | 3959.58 | 6786.27 | 6674.37 | 6735.45 | 6801.60 | 6969.60 | 7105.03 | 7702.86 | 8340.40 |
| **QwQ-32B** | | | | | | | | | |
| Tokens | 5523.47 | 5563.86 | 5673.22 | 6040.58 | 6337.92 | 6921.26 | 7511.89 | 8164.05 | 8345.72 |

## E.2 Generalization on More Model Backbones

We add experiments on another kind of LRM DeepSeek-R1-Distill-Llama-8B [2]. All of our findings also hold on this model, and we will add more results in our paper. We illustrate some datapoints of our main observations on DeepSeek-R1-Distill-Llama-8B [2]. We report the linear probing results in Table 7, Table 8, and Table 9. We also report the activation steering results in Table 10. Interestingly, DeepSeek-R1-Distill-Llama-8B is more sensitive to the steering strength $\lambda$.

Table 7: Prediction on middle layers.

| Layer | 25 | 26 | 27 | 28 | 29 | 30 |
|---|---|---|---|---|---|---|
| R | 0.8481 | 0.8467 | 0.8442 | 0.8464 | 0.8423 | 0.8430 |

Table 8: Cosine similarity between pre-allocated vectors on layer 31.

| | r2←1 | r3←1 | r4←1 | r5←1 |
|---|---|---|---|---|
| r2←1 | 1.00 | 0.98 | 0.98 | 0.90 |
| r3←1 | 0.98 | 1.00 | 1.00 | 0.93 |
| r4←1 | 0.98 | 1.00 | 1.00 | 0.94 |
| r5←1 | 0.90 | 0.93 | 0.94 | 1.00 |

Table 9: Mean cosine similarities on middle layers.

| Layer | 25 | 26 | 27 | 28 | 29 | 30 |
|---|---|---|---|---|---|---|
| R | 0.9681 | 0.9661 | 0.8873 | 0.9620 | 0.9660 | 0.9124 |

Table 10: Activation Steering results across datasets.

| Dataset | -0.02 | -0.015 | -0.01 | -0.005 | 0 | 0.005 | 0.01 | 0.015 | 0.02 |
|---|---|---|---|---|---|---|---|---|---|
| **MATH500** | | | | | | | | | |
| Tokens | 3042.42 | 3391.56 | 3312.67 | 3478.20 | 3457.01 | 3484.90 | 3530.19 | 3702.91 | 3654.19 |
| Accuracy | 75.18 | 80.00 | 81.78 | 83.55 | 82.80 | 83.38 | 83.98 | 84.88 | 84.48 |
| **AIME2024** | | | | | | | | | |
| Tokens | 10366.34 | 10910.79 | 10458.87 | 10750.82 | 11049.53 | 10774.68 | 10888.95 | 11553.91 | 11622.77 |
| Accuracy | 26.67 | 32.08 | 34.78 | 38.75 | 38.96 | 38.75 | 37.08 | 42.50 | 41.67 |
| **OlympiadBench** | | | | | | | | | |
| Tokens | 6926.34 | 7174.21 | 7254.21 | 7265.71 | 7234.64 | 7338.05 | 7522.05 | 7615.02 | 7345.39 |
| Accuracy | 43.69 | 48.24 | 49.94 | 50.13 | 49.76 | 49.59 | 50.41 | 51.20 | 49.30 |

The header above the columns -0.02 through 0.02 is $\lambda$.

### E.2.1 Case Study

**Question:** Compute: $1 - 2 + 3 - 4 + 5 - \cdots + 99 - 100$.

---

*w/o* **Steering:** `<think>`Okay, so I have this problem to compute the sum: 1 - 2 + 3 - 4 + 5 - 6 + ... + 99 - 100. Hmm, that looks like an alternating series where the signs alternate between positive and negative. Let me try to figure this out step by step. First, let me write out the series to understand the pattern better: 1 - 2 + 3 - 4 + 5 - 6 + ... + 99 - 100. I notice that this series alternates between adding an odd number and subtracting the next even number. So, each pair of terms is like (1 - 2), (3 - 4), (5 - 6), and so on. Maybe I can group them into these pairs and see what each pair adds up to. Let me compute a few pairs: 1 - 2 = -1, 3 - 4 = -1, 5 - 6 = -1, 7 - 8 = -1, ... It seems like each pair of terms adds up to -1. That's interesting. So, if I can figure out how many such pairs there are, I can multiply that number by -1 to get the total sum. Since the series starts at 1 and ends at 100, that's a total of 100 numbers. If I'm pairing them two by two, the number of pairs should be 100 divided by 2, which is 50. So, there are 50 pairs.
$\cdots$
That is, for each term, if n is odd, we add it, and if n is even, we subtract it. So, the general term is $(-1)^{(n+1)} * n$. Maybe I can use a formula for such alternating series. I remember that the sum of an alternating series can sometimes be calculated using arithmetic series formulas or by grouping terms. Wait, another approach: Maybe I can separate the series into two separate sums: the sum of the odd terms and the sum of the even terms. Let's see: The series is 1 + 3 + 5 + ... + 99 minus (2 + 4 + 6 + ... + 100). So, if I can compute both of these sums separately, I can subtract them to get the result. First, the sum of the odd numbers from 1 to 99. How many terms are there? Since we're going up to 99, which is the 50th odd number (since 99 = 2*50 - 1). So, the number of terms is 50.
$\cdots$
Wait, another way: Let's think about the series as telescoping. But I don't think it's telescoping because there's no cancellation beyond the immediate pairs. Each term cancels only part of the next term, but in this case, each pair is independent. Alternatively, maybe writing the series as: (1 - 2) + (3 - 4) + ... + (99 - 100). Each parenthesis is -1, and there are 50 of them. So, 50*(-1) = -50. Yep, same result. I think I've approached this problem in multiple ways: pairing terms, separating into odd and even sums, using the formula for alternating series, and each time I get -50. So, it seems pretty solid. Just to recap all the methods: 1. Pairing: Each pair (1-2), (3-4), etc., is -1, 50 pairs total, sum is -50. 2. Separating into odds and evens: Sum of odds is 2500, sum of evens is 2550, difference is -50. 3. Using the formula for the sum of an alternating series of consecutive integers with even terms: -n/2, which gives -50. All methods lead to the same conclusion, so I think it's safe to say the answer is -50. **Final Answer** The value of the series is \boxed{-50}`</think>`. (2253 tokens)

---

*w* **Steering:** `<think>`First, I observe that the sequence alternates between addition and subtraction of consecutive integers. To simplify the calculation, I can group the terms in pairs: (1 - 2), (3 - 4), (5 - 6), and so on, up to (99 - 100). Each pair results in -1. Since there are 100 numbers in total, there are 50 such pairs. Multiplying the number of pairs by each result gives 50 times -1, which equals -50.`</think>` (113 tokens)

Figure 23: Case study on R1-Distill-Qwen-32B. The model generates the correct answer both with (*i.e.,* $w$) and without (*i.e.,* $w/o$) steering, but steering significantly reduces the reasoning token number.

**Question:** What is the correct answer to this question: Kant's humanity formulation of the categorical imperative makes it impermissible for us to, he argues, Choices: (A) sell one of our teeth. (B) engage in prostitution. (C) both A and B (D) Neither A nor B Only put the option letter in the box, e.g.

boxedA. There is only one correct answer, .

---

$w/o$ **Steering:** `<think>` Okay, so I need to figure out what Kant's humanity formulation says about selling one of our teeth or engaging in prostitution. Let me start by recalling what the humanity formulation is. From what I remember, Kant's categorical imperative has several formulations, and one of them is the humanity formulation. The key idea here is that we should treat humanity, whether in oneself or in others, always as an end and never merely as a means to an end. So, the question is asking which actions Kant would consider impermissible under this principle. The options are selling one's tooth, prostitution, both, or neither. First, let me break down each option. Selling a tooth: that's essentially selling part of your body. Now, in the context of Kant's ethics, using oneself as a means to an end might be problematic.

· · ·

Wait, but what about the actual examples? I remember that Kant specifically mentions prostitution in his works, right? He considered it immoral because it treats the person as an object for sexual use, which is a means to an end. Similarly, selling body parts might be similar. But selling a tooth is a physical part, whereas prostitution is an action. However, in both cases, the person is using their body for monetary gain, possibly reducing themselves to a means. But maybe Kant would say that selling a tooth is permissible if it's for a good reason, like saving a life, but not sure. Wait, the formulation is about treating humanity as an end. So any action that uses a person (including oneself) merely as a means would be wrong. So selling your tooth would involve using your body as a means to get money, but does that mean you're treating yourself as a mere means? Maybe. Because you're using your body as a commodity, rather than valuing it as an end in itself. Alternatively, maybe Kant would distinguish between selling a tooth (a part) and selling services (like prostitution). But I think both involve commodifying the body. So perhaps both are impermissible. Thus, the answer would be (C) both A and B.

· · ·

Alternatively, maybe he only opposes prostitution, but I think both. Let me think again. Another angle: the humanity formulation requires that in any action, you never use a person merely as a means. So if you sell your tooth, you are using yourself as a means (to get money), but you're also an end. However, if you are selling a tooth, maybe you are treating yourself as an end in some way. Wait, perhaps the distinction is whether you are using your body as a mere means. If you sell a tooth, you are using that part of your body as a tool, but does that mean you are treating your humanity as a means? The humanity is your rational nature. So maybe selling a tooth doesn't directly violate that, as it's a physical part. Alternatively, Kant might argue that the body is part of what makes a person, so using it as a commodity reduces oneself. So perhaps both are wrong. Since I'm a bit torn here, but I think the standard interpretation is that Kant would say both are impermissible. Hence, answer (C). `</think>`. (1610 tokens)

---

$w$ **Steering:** `<think>` "Okay, so I need to figure out what the answer is here. The question is about Kant's humanity formulation of the categorical imperative and what he says is impermissible. The options are selling a tooth, prostitution, both, or neither. First, I remember that Kant's categorical imperative has different formulations.

· · ·

So both selling a tooth (if it's your own) and prostitution would be impermissible because both involve treating humanity (either your own or someone else's) as a means. So the answer should be C, both A and B? But wait, maybe selling a tooth isn't considered the same as prostitution? Like, a tooth is part of your body but maybe it's not using humanity as an end? Hmm, but Kant might say that even your own body shouldn't be treated merely as a resource to exploit for gain. So selling a tooth could still be wrong because you're using your own body as a means to an end (making money) rather than respecting your own humanity as an end in itself. So both A and B are impermissible. Therefore, the answer should be C. " `</think>` (418 tokens)

Figure 24: Case study on QwQ-32B. The model generates the correct answer both with (*i.e.,* $w$) and without (*i.e.,* $w/o$) steering, but steering significantly reduces the reasoning token number.

# F  Broader Impacts

This paper aims to investigate whether LRMs pre-plan their reasoning strength within their activation space, and how such planning is encoded with pre-allocated direction vectors. Our study contributes to a deeper understanding of LRMs within the LLM research community.

However, our findings may also pose potential risks. For instance, malicious methods could exploit the discovered property that reasoning length can be manipulated through the model's internal activations to implant backdoor attacks. Such attacks might trigger excessively long chains of thought under specific conditions, thereby significantly slowing down model execution.

