# OpenReview forum: "On Reasoning Strength Planning in Large Reasoning Models"
_NeurIPS.cc/2025/Conference — NeurIPS 2025 poster_

### Official Review · Reviewer_ZwPk · 2025-06-21

**Clarity:** 3
**Significance:** 3
**Originality:** 3
**Rating:** 5
**Confidence:** 4

**Summary:**

This paper uses techniques from mechanistic interpretation to identify a directional vector which corresponds to the number of thinking tokens used by Qwen reasoning models prior to reaching a final answer. They show that this direction vector exists prior to the beginning of token generation, and that changing it effects the number of tokens generated. They also show improvements in performance on math olympiad benchmarks simply by increasing the activation of this vector a small amount.

**Questions:**

1. How do we know that these results are not an artefact of training Qwen to solve math olympiad problems? I would strongly encourage you to show results on another reasoning benchmark, since the three shown are fundamentally the same domain. A logical reasoning benchmark without mathematics would be ideal. Including another model would be nice, but less important.

2. Can you get a similar effect simply by prompting the model to use more thinking tokens? Does doing so change the activations in this vector?

3. Under what conditions does more reasoning lead to better responses? Can you tell when using this vector would be helpful versus harmful?

**Ethical Concerns:**

["NO or VERY MINOR ethics concerns only"]

**Final Justification:**

**Quality:**

The authors addressed my concerns about generality in showing an additional benchmark. I also concede from discussion that choosing the right lambda parameter does not need to be in the scope of this work. The authors have agreed to clarify the terminology in the final paper.

**Clarity:**

I would still be in favor of further effort to clean up the text, but the text is clear enough to communicate the results.

**Significance/Originality:**

The paper does make a useful contribution.

**Limitations:**

Yes, though I think some of the weaknesses should be addressed in more detail as described above.

**Quality:**

3

**Strengths And Weaknesses:**

**Quality:**

*Strengths:* The results seem to be technically sound, and the methods are logical. The finding that a single dimension largely controls the length of LLM generations is likely to be true given the evidence provided.

*Weaknesses:* As the authors acknowledge, the paper is narrow in focus, testing only Qwen models and evaluating performance only on math olympiad benchmarks (with a bit of physics). This raises the important question: are the results a training artefact from how the Qwen models were trained for math olympiad puzzles? I do think this is a serious concern, particularly when the claims made refer to "planning" which may just be a recognizing of a certain type of problem from pre-training. I would strongly encourage showing results on another type of reasoning benchmark.

Two other methodological concerns:

- The results on L222 about "moderate" steering seem difficult to use in practice. How would we predict how much steering is beneficial versus harmful? Without a principled reason to use this amount of steering, the results in Table 1 are an  unfair comparison.

- The paper consistently claims that allocating more or less tokens is a planning strategy related to reasoning strength. However, I am not convinced by the connection made between "length" and "strength". Other works have found that models can overthink (e.g. https://arxiv.org/abs/2502.07266) and have lower performance with more tokens.

**Clarity:**

*Strengths:* The submission is generally well-organized. The text is clearly written.

*Weaknesses:* The organization of the sections is a bit confusing. There are a number of sections which jump back and forth between methods and results, but only one section called results. The placement of the figures is also annoyingly far from the references in the text, though less of this blame falls on the authors.

**Significance/Originality:**

*Strengths:* The paper makes a useful contribution to controlling the behavior of reasoning LLMs. The work is original to my knowledge and could be useful for improving training methods in future models.

*Weaknesses:* The methods themselves are not original, but this is not the intention of the paper.

---

> ### Author Rebuttal · Authors · 2025-07-31
>
> Thanks for your positive feedback, and we sincerely appreciate your valuable suggestions. We respond to your questions as follows.
>
> > **Results on another type of reasoning benchmark.**
>
> Thanks for your concerns. We extend our experiments in Table 1 to another complex logic reasoning task GPQA diamond [1] for better demonstrating the generalization capabilities. We show the results on GPQA diamond in the following table:
>
> **Table 1: The effect of steering and performance on GPQA diamond**
>
>
> | λ                    |          | -0.2    | -0.15   | -0.1    | -0.05   | 0       | 0.05    | 0.1     | 0.15    | 0.2     |
> | -------------------- | -------- | ------- | ------- | ------- | ------- | ------- | ------- | ------- | ------- | ------- |
> | R1-Distill-Qwen-1.5B | Tokens   | 959.44  | 2930.29 | 5659.00 | 6677.72 | 7428.22 | 7690.48 | 7902.20 | 7805.64 | 7784.97 |
> |                      | Accuracy | 8.01    | 6.05    | 8.20    | 12.30   | 13.86   | 16.80   | 17.58   | 16.80   | 19.53   |
> | R1-Distill-Qwen-7B   | Tokens   | 1022.10 | 2755.88 | 5250.92 | 6243.68 | 6998.01 | 7326.04 | 7875.52 | 8103.89 | 8380.32 |
> |                      | Accuracy | 5.66    | 12.30   | 22.27   | 28.13   | 33.92   | 36.13   | 40.23   | 39.26   | 40.33   |
> | R1-Distill-Qwen-14B  | Tokens   | 886.26  | 2289.21 | 3910.04 | 6035.36 | 6365.84 | 6167.72 | 5927.13 | 5463.18 | 5137.19 |
> |                      | Accuracy | 15.42   | 17.97   | 23.24   | 44.34   | 50.78   | 51.95   | 51.56   | 51.76   | 50.20   |
> | R1-Distill-Qwen-32B  | Tokens   | 266.07  | 296.68  | 1612.85 | 4696.14 | 5881.31 | 6391.68 | 6809.91 | 7128.37 | 7274.20 |
> |                      | Accuracy | 17.38   | 19.33   | 19.14   | 44.73   | 55.76   | 56.83   | 59.77   | 59.38   | 54.30   |
> | QwQ-32B              | Tokens   | 6490.33 | 6826.64 | 7368.09 | 7472.53 | 8087.62 | 8811.63 | 8763.01 | 9161.22 | 9862.67 |
> |                      | Accuracy | 48.43   | 58.79   | 59.38   | 59.76   | 60.55   | 60.96   | 60.16   | 59.38   | 55.86   |
>
> > **The choice of steering strength**
>
> Thanks for your concern about the choice of steering strength. Actually, in most cases, we found λ=0.05 generally works fine on all the datasets. In few cases, λ =0.1 works a little bit better than λ = 0.05. Therefore, we would suggest using a default λ = 0.05, since it generally benefits. Additionally, for better predicting how much steering is beneficial versus harmful, we can observe the performance on the validation set for choosing the proper λ. Moreover, since the main contribution of this paper is revealing the findings in reasoning strength planning, we just choose the best λ for testing the effectiveness of steering with such pre-allocation vectors. I believe it would also be important to discuss how to make our findings into practical methods, such as more fine-grained controls [2], although this is not our main contribution.
>
> > **Questions about reasoning strength.**
>
> Thanks for your questions. In the first sentence of the abstract, we define the reasoning strength as the number of reasoning tokens. Therefore, it is generally the same as the reasoning length you have mentioned. I also agree with the insight that overthink can lead to decreased performance, as shown in the previous work [3, 4]. Additionally, our findings also support such insights. As shown in Figure 6, the performance rises and then drops as the reasoning token number increases. This indicates that adding reasoning strengths moderately can improve the performance, while overthink can lead to decreased performance.
>
>
>
> > **Questions**
>
> - Question 1. We have added experiments on GPQA diamond as shown above.
>
> - Question 2. We test whether we can yield more reasoning tokens by simply adding one sentence: "Please reason with more tokens." As shown in this table, naively prompting with more reasoning tokens does not consistently bring more reasoning tokens, as we even observe some reasoning token number drop in 1.5B and 14B models. Such results are consistent with previous findings that LRMs suffer from prompting with token numbers [5].
>
>   **Table 2: The effect of prompting LRMs to reason with more tokens**
>
>   |                      | Original | + Prompting |
>   | -------------------- | -------- | ----------- |
>   | R1-Distill-Qwen-1.5B | 4899.34  | 4421.98     |
>   | R1-Distill-Qwen-7B   | 3828.50  | 4004.16     |
>   | R1-Distill-Qwen-14B  | 3495.39  | 3091.04     |
>   | R1-Distill-Qwen-32B  | 3430.73  | 3453.11     |
>   | QwQ-32B              | 4222.67  | 4718.66     |
>
> - Question 3. Generally, as we have responded to the weakness 2, for increasing the performance, we can generally use a default λ=0.05. For reducing the token numbers, we can steer with negative strength λ on easy questions (in practice, we can train another detector for detecting the difficulty of questions). However, if the steering strength is too big, it will cause harmful results, such as decoding errors or overlong responses.
>
> [1] GPQA: A Graduate-Level Google-Proof Q&A Benchmark
>
> [2] SEAL: Steerable Reasoning Calibration of Large Language Models for Free. 2025.4.
>
> [3] When More is Less: Understanding Chain-of-Thought Length in LLMs
>
> [4] Do NOT Think That Much for 2+3=? On the Overthinking of o1-Like LLMs
>
> [5] L1: Controlling How Long A Reasoning Model Thinks With Reinforcement Learning. Arxiv 2025.3.

---

> > ### Comment · Reviewer_ZwPk · 2025-08-01
> > **Thank you**
> >
> > Thank you for your helpful response.
> >
> > The additional results on GPQA are useful in showing generalisation, thank you.
> >
> > The choice of steering strength comments are somewhat less convincing, as the response to a different reviewer showed the Llama-based model have an optimal \lambda of ~0.005 rather than 0.05. However, I do agree that this is not essential for this paper to solve.
> >
> > I do still think a clearer distinction should be made between "reasoning strength" and "number of tokens". I see that you have equated the two, but disagree with the terminology. The example I gave of having lower performance with more tokens is meant to indicate that higher "strength" as measured by performance does not correspond to higher "strength" as measured by token count, indicating a mismatch in the terminology.
> >
> > I'm glad to know that prompting doesn't easily replicate these results.

---

> > > ### Author Response · Authors · 2025-08-03
> > > **Thanks for your responses**
> > >
> > > We sincerely appreciate your valuable and constructive comments on our paper. We are also grateful that you have carefully reviewed the feedback from other reviewers as well as our responses.
> > >
> > > Your suggestions have greatly helped us refine our work. Regarding the issue you raised about terminology, we will make a clearer distinction in the revised paper.
> > >
> > > We sincerely hope that our efforts and the improvements made during the rebuttal process will be taken into consideration. If our responses have addressed most of your questions and you acknowledge the efforts we have put into refining the work, we would greatly appreciate it if you could consider adjusting your evaluation accordingly.
> > >
> > > Thanks again for your time, expertise, and invaluable suggestions throughout this process.

---

> > > > ### Comment · Reviewer_ZwPk · 2025-08-04
> > > > **Thank you**
> > > >
> > > > I'm happy to recommend this paper for acceptance, provided they follow through with clarifying the terminology. The remaining new results have clarified my other concerns, and I believe the paper will be useful for future research and development.
> > > >
> > > > (I suppose it will vary by reviewer, but you don't need to be this complimentary if your results are sound.)

---

> > > > > ### Author Response · Authors · 2025-08-05
> > > > > **Thanks**
> > > > >
> > > > > We sincerely appreciate your time and recommendation. We would consider your valuable suggestions in our latest paper.

---

### Official Review · Reviewer_1xAp · 2025-06-27

**Clarity:** 4
**Significance:** 3
**Originality:** 3
**Rating:** 5
**Confidence:** 4

**Summary:**

This paper investigates the underlying mechanisms of reasoning strength planning in Large Reasoning Models (LRMs) and reveals that these models pre-allocate reasoning strength before generation. The authors employ linear probing and activation steering techniques to demonstrate that: (1) LRMs can predict the required reasoning token count solely from question activations before generation; (2) LRMs encode reasoning strength through pre-allocated directional vectors in activation space, where vector magnitude modulates reasoning strength; and (3) these vectors causally control reasoning by affecting the logits of the end-of-reasoning token. At the same time, it put forwards two important applications of this finding on solving overthinking and improving reasoning efficiency. Overall, this paper is well structured and presents novel insights that advance our understanding of how LRMs manage computational resources during reasoning tasks. The findings have both theoretical significance and practical implications for optimizing reasoning performance in large language models.

**Questions:**

1. For results shown in figure 7, the predicted token length $y ̂^{(l)}$ exhibit high volatility across different layers, with some predictions even showing negative values. Why This appears inconsistent with the findings in Figure 2, where deeper layers theoretically should yield more accurate and stable predictions. For example, in figure 7(a), layers 20-25 demonstrate high Spearman correlation coefficients in Figure 2(a), yet the predicted token lengths in these deeper layers are relatively small and sometimes negative. This discrepancy warrants further explanation regarding why high correlation does not translate to more reasonable absolute predictions.

2. For figure 8, why not include results from larger models 32B? Cause Based on previous findings, larger models appear to better align with the hypothesis. For instance, in Figure 2, the Spearman correlation coefficients for the 32B model are consistently high across most layers, and in Figure 5, the L2 norm changes are more pronounced for the 32B model.

3. Regarding the results in Figure 10 using the MATH dataset, QwQ-32B shows significantly less reduction in token length after steering compared to R1-Distill-Qwen-32B. Is this difference due to the use of a different negative steering strength $λ$, or are there other factors?

4. In section 5.2, for the experiments shown in figure 10, how to define the "proper negative steering strength $λ$ "?

**Ethical Concerns:**

["NO or VERY MINOR ethics concerns only"]

**Final Justification:**

The detailed explanation solved most of my concerns. However, as indicated by Reviewer qGsi and my previous comments, the generalization of this method to other reasoning tasks seems to be limited (MATH). Although result of GPQA is provided, it is focused more on scientific knowledge. As a general LRM approach, I prefer data coding generation tasks such as LiveCodeBench to validate the reasoning ability.

After discussion, LiveCodeBench is also provided and the result is validated. The generalization ability is well demonstrated thus I recommend this paper to be accepted.

I do appreciate the effort of the authors to conduct extensive experiments and I believe that this paper will be helpful to the domain of LRM.

**Limitations:**

yes

**Quality:**

4

**Strengths And Weaknesses:**

Strength:

•	This paper fills the gap that provide the first systematic investigation into the internal mechanisms of reasoning strength planning in LRMs, addressing an important gap in understanding how these models automatically allocate computational resources. The findings are well presented and the experimental designs are clearly articulated.

•	This paper applies solid analysis approach through the combination of linear probing and activation steering, which provides compelling evidence for the existence of pre-allocated reasoning strength vectors. In addition, the causal intervention experiments using activation steering effectively demonstrate the functional role of these vectors.

•	This paper provides comprehensive analysis including compariing the actiavtion directions and magnitudes across different difficulty levels and invetiagting on the effect of activation magnitude steering and enhancement to the reasoning token length.

•	This paper provides insights of concrete applications including overthinking detection and efficient inference, showing the potential real-world value of these findings.

Weakness:

•	Firstly, domain limitation, the analysis is predominantly conducted on mathematical reasoning tasks. While as stated in introduction, coding generation tasks also require reasoning, the author may consider adding experiments in section 4 with coding generation dataset. At the same time, the author may consider adding experiments with reasoning VLMs to test on different scenarios other than LLMs.

•	Secondly, insufficient analysis of counter cases, the paper doesn’t thoroughly analyze the cases where the model doesn’t follow the founded relationships between reasoning token lengths and activation magnitude. For example, as noted in the introduction, post-training methods such as SFT and RL can affect reasoning strength. The authors should consider incorporating counter-case analyses, such as comparative experiments between pre-trained and post-trained models, to strengthen their claims.

•	Thirdly, limited baseline comparisons and evaluation. The author may consider comparing with alternative methods for reasoning length control while maintaining accuracy. Moreover, in section 5.2, the authors should expand their evaluation beyond easy tasks to include harder tasks, examining whether activation steering causes significant degradation in accuracy performance on harder tasks.

---

> ### Author Rebuttal · Authors · 2025-07-31
>
> We sincerely appreciate your detailed review and positive feedback for our work! We address your questions as follows.
>
> > **Domain limitation**
>
> Thanks for your suggestions.
>
> - **Generalization capabilities in other domains.** We discussed the generalization capabilities of our findings on other general domains beyond MATH, such as AlpacaEval and MMLU in Section 5, where our findings of reasoning strength prediction and strength control also hold on these two datasets. Due to the evaluation difficulty of code tasks on LRMs [1], following previous works [2, 3], we add one more experiment on the complex logic reasoning task GPQA diamond for better demonstrating the generalization capabilities.
>
> **Table 1: The effect of steering and performance on GPQA diamond**
>
> |λ||-0.2|-0.15|-0.1|-0.05|0|0.05|0.1|0.15|0.2|
> |-|-|-|-|-|-|-|-|-|-|-|
> |R1-Distill-Qwen-1.5B|Tokens|959.44|2930.29|5659.00|6677.72|7428.22| 7690.48|7902.20|7805.64|7784.97|
> ||Accuracy|8.01|6.05|8.20|12.30|13.86|16.80|17.58|16.80|19.53|
> |R1-Distill-Qwen-7B|Tokens|1022.10|2755.88|5250.92|6243.68|6998.01|7326.04|7875.52|8103.89|8380.32|
> ||Accuracy|5.66|12.30|22.27|28.13|33.92|36.13|40.23|39.26|40.33|
> |R1-Distill-Qwen-14B|Tokens|886.26|2289.21|3910.04|6035.36|6365.84| 6167.72|5927.13|5463.18|5137.19|
> ||Accuracy|15.42|17.97|23.24|44.34|50.78|51.95|51.56|51.76|50.20|
> |R1-Distill-Qwen-32B|Tokens|266.07|296.68|1612.85|4696.14|5881.31 | 6391.68|6809.91|7128.37|7274.20|
> ||Accuracy|17.38|19.33|19.14|44.73|55.76|56.83|59.77|59.38|54.30|
> |QwQ-32B|Tokens|6490.33|6826.64|7368.09|7472.53|8087.62|8811.63|8763.01|9161.22|9862.67|
> ||Accuracy|48.43|58.79|59.38|59.76|60.55|60.96|60.16|59.38|55.86|
>
> - **Exploration on VLMs.** Thanks for your valuable suggestions. We believe this is a promising research direction and would be another good research work. Extending our findings requires extending our methodologies to new modalities, datasets, and benchmarks. This requires a large amount of computational resources, since this research is very computation-consuming (i.e., we spent more than 100 A100 GPU days for the whole experiment) and experiments on VLMs would be even more computation-consuming. Adding experiments would be our future work, given the time limit of the rebuttal and the lower maturity of current reasoning VLMs.
>
> > **Counter cases.**
> - Thanks for your suggestion, but I probably may not fully understand your question. Although our findings are consistent in 90% of the cases across models and datasets, we do find some cases where the model doesn’t follow the found relationships between reasoning token lengths and activation magnitude. As shown in Figure 18.a in Appendix C.2, on R1-Distill-Qwen-14B, the reasoning strength slightly decreases under a positive steering strength. We hypothesize this is probably due to some inner differences between R1-Distill-Qwen-14B and other models, which is worth exploring in our future work. We may discuss this later in our discussion period if my understanding is inaccurate.
>
>
>
> > **Limited baseline comparisons and evaluation**
>
> Thanks for your suggestions.
>
> - **Comparison with other baselines.** Thanks for your suggestions. Existing length controlling methods typically require training on additional corpora [1, 2]. Since our paper does not contain any training parts, it is unfair to compare it with these methods. We also add one more training-free method, which multiplies the logits of the end-of-reasoning token with a factor γ (i.e., logit_new = γ*logit_old). We report the results on MATH500 in Table 2. We find that naively changing the logits will lead to unstable manners, either leading to overly increased answer token number (i.e., γ = 2 on R1-Distill-Qwen-7B) or easily leading to decoding errors (i.e., γ = 0.8). This further indicates the effectiveness of steering.
>
> **Table 2: Effect of changing the logits with γ **
>
> |                    | γ                   | Reasoning Token Number | Answer Token Number | Accuracy |
> | --------------------| -------------------- | ------- | ------------ | ------- |
> | R1-Distill-Qwen-1.5B | 2    | 4445.93 | 376.89 | 82.00 |
> |                      | 1    | 4316.95 | 372.12 | 82.78 |
> |                      | 0.8  | 16159.28 | 2.0 | 0.15 |
> | R1-Distill-Qwen-7B | 2 | 1013.48 | 5795.13 | 56.20 |
> |                      | 1 | 3392.95 | 386.03 | 0.9218 |
> |  | 0.8 | 11309.18 | 2 | 0.0 |
> | R1-Distill-Qwen-14B | 2    | 3267.30 | 387.06 | 94.35 |
> |                      | 1    | 3148.91 | 391.72 | 93.37 |
> |                      | 0.8  | 15913.95               | 2.0                 | 0.15     |
> | R1-Distill-Qwen-32B  | 2    | 3038.15 | 393.72 | 95.10 |
> |                      | 1    | 3029.32 | 395.75 | 94.05 |
> |                      | 0.8  | 6963.46                | 2.0                 | 0.0      |
> | QwQ-32B              | 2    | 3623.64 | 423.12 | 95.45 |
> |                      | 1 | 3690.35 | 436.25 | 95.90 |
> |                      | 0.8 | 16256.47 | 1.0 | 0.0 |
>
> - **Expansion to harder tasks.** As we have detailed in Section 5.2, our main goal is to test whether the LRMs indeed largely overthink on easy questions and how to make them think more efficiently on these questions. As shown in Figure 6, we can still reduce a certain number of reasoning tokens while maintaining the performance on harder questions, but the degree is much smaller than that on easy questions. Such findings are consistent with existing works that LRMs tend to overthink more on easy questions [4].
>
>
>
> > **Questions**
>
> - **High volatility across different layers.** Thanks for your question. We hypothesize the volatility may come from the fact that we find the main direction for reasoning strength planning, while there may exist several sub-directions for more fine-grained reasoning strength planning, as previous works have observed in the field of LLM safety [5, 6].
> - **Results for QwQ-32B.** Thanks for your question. We observe similar patterns on QwQ-32B, and we will put the results in the appendix later.
> - **Less reduction in token length for QwQ-32B.** We hypothesize that this difference may come from the training difference between the Distilled-R1 series and QwQ-32B. Distilled-R1 series are trained with supervised finetuning (SFT) while QwQ-32B is trained with reinforcement learning (RL). We hypothesize that SFT may introduce more overthink patterns since they are trained with pre-collected data, while RL may make the reasoning strength planning more accurate since the training data is generated by the model itself. Therefore, QwQ-32B may be more sensitive to the changes in their reasoning token numbers.
> - **Proper negative steering strength.** Sorry for making this confusing. We choose the steering strength that can largely maintain the performance while reducing the reasoning token numbers as much as possible. This is what we call proper steering strength. We will make this clearer in our latest paper.
>
> [1] s1: Simple test-time scaling. Arxiv 2025.1.
>
> [2] L1: Controlling How Long A Reasoning Model Thinks With Reinforcement Learning. Arxiv 2025.3.
>
> [3] Open-reasoner-zero: An open source approach to scaling up reinforcement learning on the base model. Arxiv.3.
>
> [4] A Survey of Efficient Reasoning for Large Reasoning Models: Language, Multimodality, and Beyond. Arxiv 2025.5.
>
> [5] The Hidden Dimensions of LLM Alignment: A Multi-Dimensional Safety Analysis. ICML 2025.
>
> [6] The Geometry of Refusal in Large Language Models: Concept Cones and Representational Independence. ICML 2025.

---

> > ### Comment · Reviewer_1xAp · 2025-08-05
> > **Thanks**
> >
> > Thanks for the response.
> >
> > The detailed explanation solved most of my concerns. However, as indicated by Reviewer qGsi and my previous comments, the generalization of this method to other reasoning tasks seems to be limited (MATH).
> >
> > I do appreciate the effort of the authors to conduct extensive experiments and I believe that this paper will be helpful to the domain of LRM.
> >
> > I keep my score in this stage.

---

> > > ### Author Response · Authors · 2025-08-05
> > > **Thanks**
> > >
> > > Thanks for your follow-up responses and acknowledgement in addressing your concerns.
> > >
> > > For the generalization on other domains, we added additional results on the general reasoning task of GPQA diamond (i.e., not a math dataset), which is acknowledged by reviewer ZwPk.
> > >
> > > We sincerely hope our improvements can be taken into consideration. If our results have addressed most of your concerns, we would appreciate it if you could adjust the evaluation.

---

> > > > ### Comment · Reviewer_ZwPk · 2025-08-05
> > > > **Note from Reviewer ZwPk**
> > > >
> > > > For what its worth, I don't think GPQA fits very well in the "reasoning" benchmark category, but it does show that the results generalize outside of mathematics. As a matter of personal taste I would choose a different benchmark that is more focused on reasoning rather than factual information.

---

> > > > ### Comment · Reviewer_1xAp · 2025-08-05
> > > > **Generalization**
> > > >
> > > > Although result of GPQA is provided, it is focused more on scientific knowledge. As a general LRM approach, I prefer data coding generation tasks such as LiveCodeBench to validate the reasoning ability. I keep this as my concern and maintain the positive score.

---

> > > > > ### Author Response · Authors · 2025-08-08
> > > > > **Thanks and Follow-up Results**
> > > > >
> > > > > Thanks for your follow-up questions.
> > > > >
> > > > > According to your suggestions, we have added two more experiments on the LiveCodeBench datasets. We evaluate the effect of steering on the two tasks: Code Execution and Code Generation. In code execution, the LRM will guess the corresponding output of the given input. And in code generation, LRMs will generate the code solution for a specific task. We report the results in Table 1 and Table 2, respectively.
> > > > >
> > > > > As shown in these tables, we found our findings also hold on the LiveCodeBench (i.e., this pre-allocation vector also affects the reasoning token numbers for code tasks). Since we found it hard to extract the generated code in a correct format for evaluation for code generation tasks when varying the response lengths [1, 2], we just report the effect on the reasoning token numbers.
> > > > >
> > > > > There is only one exception on R1-Distill-Qwen-1.5B, where reducing the reasoning token number even brings the performance gain. We carefully analyze this model and find that when generating overlong responses, it tends to forget to follow the original instruction, and sometimes the results get cut off. Therefore, reducing the reasoning token number helps it avoid such bad issues like cut off, and increasing it will have the opposite effect.
> > > > >
> > > > > We hope our new results can address your issues about the generalization capabilities in the general reasoning domain.
> > > > >
> > > > > **Table 1: The effect of steering and performance on LiveCodeBench (Code Execution)**
> > > > >
> > > > > | λ                    |          | -0.2    | -0.15   | -0.1    | -0.05   | 0       | 0.05    | 0.1     | 0.15    | 0.2      |
> > > > > | -------------------- | -------- | ------- | ------- | ------- | ------- | ------- | ------- | ------- | ------- | -------- |
> > > > > | R1-Distill-Qwen-1.5B | Tokens   | 1971.90 | 2569.96 | 3112.51 | 3942.85 | 4682.46 | 5856.20 | 7120.98 | 8660.37 | 10482.23 |
> > > > > |                      | Accuracy | 26.30   | 23.02   | 18.32   | 13.20   | 8.72    | 5.48    | 2.30    | 1.41    | 0.84     |
> > > > > | R1-Distill-Qwen-7B   | Tokens   | 281.09  | 358.46  | 678.20  | 1209.86 | 1625.67 | 2198.33 | 2942.55 | 3701.05 | 4688.05  |
> > > > > |                      | Accuracy | 61.85   | 66.75   | 72.65   | 79.33   | 79.23   | 79.30   | 74.63   | 80.17   | 80.85    |
> > > > > | R1-Distill-Qwen-14B  | Tokens   | 346.57  | 461.33  | 929.89  | 1267.48 | 1428.35 | 1562.22 | 1745.25 | 1966.76 | 2197.66  |
> > > > > |                      | Accuracy | 68.42   | 74.01   | 82.05   | 91.54   | 91.03   | 91.28   | 89.35   | 89.35   | 89.67    |
> > > > > | R1-Distill-Qwen-32B  | Tokens   | 230.60  | 345.78  | 501.81  | 827.44  | 1201.67 | 1682.29 | 2317.60 | 2916.74 | 3465.59  |
> > > > > |                      | Accuracy | 52.14   | 56.89   | 58.51   | 68.58   | 79.54   | 86.69   | 90.03   | 93.63   | 86.01    |
> > > > > | QwQ-32B              | Tokens   | 1053.54 | 1151.99 | 1284.14 | 1439.79 | 1682.31 | 1895.04 | 2202.13 | 2521.79 | 2927.20  |
> > > > > |                      | Accuracy | 93.74   | 94.99   | 96.45   | 98.38   | 99.06   | 99.27   | 99.27   | 98.64   | 98.49    |
> > > > >
> > > > > **Table 2: The effect of steering and performance on LiveCodeBench (Code Generation)**
> > > > >
> > > > > | λ                    |        | -0.2    | -0.15   | -0.1    | -0.05   | 0        | 0.05     | 0.1      | 0.15     | 0.2      |
> > > > > | -------------------- | ------ | ------- | ------- | ------- | ------- | -------- | -------- | -------- | -------- | -------- |
> > > > > | R1-Distill-Qwen-1.5B | Tokens | 9176.89 | 9548.69 | 9915.46 | 9999.58 | 10422.80 | 10744.27 | 10963.86 | 11083.30 | 11070.14 |
> > > > > | R1-Distill-Qwen-7B   | Tokens | 1905.08 | 6409.04 | 8333.01 | 8747.09 | 8964.33  | 9194.85  | 9549.61  | 9635.65  | 10233.36 |
> > > > > | R1-Distill-Qwen-14B  | Tokens | 6732.97 | 6987.67 | 7578.29 | 7310.26 | 7299.31  | 7238.23  | 7008.38  | 6797.31  | 6494.35  |
> > > > > | R1-Distill-Qwen-32B  | Tokens | 3959.58 | 6786.27 | 6674.37 | 6735.45 | 6801.60  | 6969.60  | 7105.03  | 7702.86  | 8340.40  |
> > > > > | QwQ-32B              | Tokens | 5523.47 | 5563.86 | 5673.22 | 6040.58 | 6337.92  | 6921.26  | 7511.89  | 8164.05  | 8345.72  |
> > > > >
> > > > > [1] s1: Simple test-time scaling. Arxiv 2025.1.
> > > > >
> > > > > [2] L1: Controlling How Long A Reasoning Model Thinks With Reinforcement Learning. Arxiv 2025.3.

---

> > > > > > ### Comment · Reviewer_1xAp · 2025-08-08
> > > > > > **Raise score**
> > > > > >
> > > > > > Thank you for your thoughtful feedback. I'm delighted that you provided the additional LiveCodeBench experiments and found the results compelling.
> > > > > > It clarified most of my major concerns and thus will raise my score.

---

> > > > > > > ### Author Response · Authors · 2025-08-08
> > > > > > > **Thanks**
> > > > > > >
> > > > > > > Thanks for all your valuable suggestions and feedback during the rebuttal phase. We sincerely appreciate your suggestions and they greatly help us refine our paper. We will include all your suggestions in our latest paper. Thanks again.

---

### Official Review · Reviewer_XAya · 2025-07-04

**Clarity:** 3
**Significance:** 2
**Originality:** 2
**Rating:** 4
**Confidence:** 4

**Summary:**

This paper studies LLMs' internal representations related to their output CoT lengths. It finds that representations right after the question encodes the CoT length to some degree using a linear probe. Further, it finds a vector direction that correlates with the length. Finally, it is shown that altering internal representation using this vector does change the resulting CoT lengths.

**Questions:**

- it is not clear which layer's representation is chosen in the experiments. For example, how is it chosen for fig6? Does each lambda value use a different layer, or it is shared?
- similar to above, how is lambda chosen for tab1 results? what are the actual values? (can be written in the table) Also the same questions for fig10.
- why fig2c has x=0 point but not the others?
- about sec5.1, does the overthink questions actually do have longer responses? how much longer in average?

**Ethical Concerns:**

["NO or VERY MINOR ethics concerns only"]

**Final Justification:**

The author rebuttal provided convincing evidences that alleviated my concerns to some degree, thus I raised my score.

**Limitations:**

yes

**Quality:**

2

**Strengths And Weaknesses:**

The paper looks into an interesting topic of CoT length. Its finding about internal representations corresponding to CoT lengths is intriguing, especially because it is measured before the CoT begins. There is also a practical application of this finding to change output CoT lengths. However, I have several concerns:

- As the authors discuss, it is possible that this internal vector might be controlling the output probability of </thought> token. Such a superficial control would be less interesting because one can directly control that token probability instead. So this can be another baseline where one simply controls this token probability by manually increasing it or decreasing it.

- Another question I have is about the strength of this correlation. How strong is spearman R 0.84? If one makes a prediction using it, what would be the average error in percentage? Knowing this will help one to get intuitive feeling about the accuracy. Another possible analysis to compare two random CoTs from the same question. How strongly they will correlate? Multiple responses sampled will have differing lengths for the same question.

- About the practical application, it seems only small performance gain can be achieved with this method. As the CoT length increase, the performance actually drops as shown in fig6. This of course limits the application of this work.

---

> ### Author Rebuttal · Authors · 2025-07-31
>
> We sincerely appreciate your suggestions and interest in our work!
>
> > **The control of token logits**
>
> Thanks for your question. We dive deeper into the logit control mechanism.
>
> - **Issues about superfacial control**. We study the logits of the end-of-reasoning token since it is the most intuitive. Following your suggestions, we further test the changes in the logits of reasoning-related tokens. We find that with a positive steering strength, the logits of complex reasoning-related tokens also increase, beyond merely decreased logits of the end-of-reasoning token </think>.
>
>   We report the changes in the logits for these reasoning-related tokens as follows in Table 1. These tokens are usually regarded as reflection patterns in the R1-style response. Increasing the logits of these tokens increases the probabilities of reflection and increases of reasoning token numbers, and decreasing them has the opposite effect. As shown in Table 1, positive steering strengths increase the logits of these tokens while negative steering strengths decrease them. This indicates the reasoning strength control is not superficial. We will add these experiments to our latest paper.
>
> **Table 1: Impact of activation steering on logits of reasoning-related tokens.**
>
> | λ       | -0.2       | -0.1        | 0.1        | 0.2        |
> | ------- | ---------- | ----------- | ---------- | ---------- |
> | Alright | -0.2033    | -0.0638     | +0.0117    | +0.0061    |
> | Hmm     | -1.1171e-4 | -4.7589e-5  | +2.5699e-5 | +7.8185e-6 |
> | Oh      | -5.1966e-8 | -2.9514e-8  | +8.5295e-8 | +2.2929e-7 |
> | Wait    | -7.3789e-9 | -1.6405e-14 | +2.8739e-8 | +9.2974e-8 |
>
> ​	This indicates that such pre-allocated vectors do not superfacially change the logits of the end-of-reasoning token </think>.
>
> - **Comparison of superfacially changing the logits of the end-of-reasoning token </think>.** To further test whether simply changing the logits of the end-of-reasoning token can achieve the same performance, we conduct experiments by multiplying the logits of the end-of-reasoning token with a factor γ (i.e., logit_new = γ*logit_old). We report the results on MATH500 in Table 2. We find that naively changing the logits will lead to unstable manners, either leading to overly increased answer token number (i.e., γ = 2 on R1-Distill-Qwen-7B) or easily leading to decoding errors (i.e., γ = 0.8).
>
>   These results further suggest that the changes in the logits of the end-of-reasoning token are the result of planning, rather than the cause of planning.
>
> **Table 2: Effect of changing the logits with γ**
>
> |                    | γ                   | Reasoning Token Number | Answer Token Number | Accuracy |
> | --------------------| -------------------- | ------- | ------------ | ------- |
> | R1-Distill-Qwen-1.5B | 2    | 4445.93 | 376.89 | 82.00 |
> |                      | 1    | 4316.95 | 372.12 | 82.78 |
> |                      | 0.8  | 16159.28 | 2.0 | 0.15 |
> | R1-Distill-Qwen-7B | 2 | 1013.48 | 5795.13 | 56.20 |
> |                      | 1 | 3392.95 | 386.03 | 0.9218 |
> |  | 0.8 | 11309.18 | 2 | 0.0 |
> | R1-Distill-Qwen-14B | 2    | 3267.30 | 387.06 | 94.35 |
> |                      | 1    | 3148.91 | 391.72 | 93.37 |
> |                      | 0.8  | 15913.95               | 2.0                 | 0.15     |
> | R1-Distill-Qwen-32B  | 2    | 3038.15 | 393.72 | 95.10 |
> |                      | 1    | 3029.32 | 395.75 | 94.05 |
> |                      | 0.8  | 6963.46                | 2.0                 | 0.0      |
> | QwQ-32B              | 2    | 3623.64 | 423.12 | 95.45 |
> |                      | 1 | 3690.35 | 436.25 | 95.90 |
> |                      | 0.8 | 16256.47 | 1.0 | 0.0 |
>
> > **Prediction accuracy & Random CoTs from the same question**
>
> - **Prediction accuracy.** We report the mean absolute error (MAE) as the accuracy metric. The predictor in Figure 1 yields MAE = 591.37 for y under 4000 tokens. Meanwhile, we add one more experiment to demonstrate that this prediction is non-trivial. We randomly permute the label y (i.e., the actual reasoning token number) and make a prediction again. The MAE under this random prediction setting is 1428.72, which is significantly higher than our prediction. We would like to highlight again that this work reveals whether and how LRMs plan the reasoning strength rather than making this prediction as accurate as possible.  It is worth exploring in the future for improving the accuracy by adding more training data, adopting a more complicated prediction model structure, and carefully tuning the hyperparameters. For example, if we merely add 500 more training datapoints from MATH500, the MAE further reduces from 591.37 to 563.42. Currently, we only have fewer than 7k training datapoints, and generating these datapoints is very time-consuming, which limits current accuracy.
>
> - **Random CoTs from the same question.** Thanks for your question. In our work, we use the setting of rollout = 8 and take the average as the final reasoning token number. We also calculate the MAE between each rollout and the average of for one rollout. The final averaged mean absolute error MAE of QwQ-32B on the MATH dataset is 602.10, indicating moderate consistency among their responses, given the average reasoning token number is over 4000.
>
> > **Small performance gain**
>
> Thanks for your questions. We would like to answer this question from the following aspects.
>
> - First, we would like to highlight that the main contribution of this paper lies in revealing whether and how LRMs plan their reasoning strength in advance, rather than designing a sophisticated method for improving the performance.
> - Second, given the naive design and the high saturation of math datasets, the improvement of steering is non-trivial [1]. Additionally, some concurrent works also suggest that steering with a more carefully designed vector can further enhance the performance, indicating that such findings can be largely improved with careful design [2].
> - Third, we add one more experiment on the general reasoning dataset GPQA diamond in Table 3. Results show that steering with the pre-allocation vector can generalize on this task and can even provide more performance gain than MATH. We attribute this to the lower saturation of such general reasoning tasks.
>
> **Table 3: The effect of steering and performance on GPQA diamond**
>
> | λ                    |          | -0.2    | -0.15   | -0.1    | -0.05   | 0       | 0.05    | 0.1     | 0.15    | 0.2     |
> | -------------------- | -------- | ------- | ------- | ------- | ------- | ------- | ------- | ------- | ------- | ------- |
> | R1-Distill-Qwen-1.5B | Tokens   | 959.44  | 2930.29 | 5659.00 | 6677.72 | 7428.22 | 7690.48 | 7902.20 | 7805.64 | 7784.97 |
> |                      | Accuracy | 8.01    | 6.05    | 8.20    | 12.30   | 13.86   | 16.80   | 17.58   | 16.80   | 19.53   |
> | R1-Distill-Qwen-7B   | Tokens   | 1022.10 | 2755.88 | 5250.92 | 6243.68 | 6998.01 | 7326.04 | 7875.52 | 8103.89 | 8380.32 |
> |                      | Accuracy | 5.66    | 12.30   | 22.27   | 28.13   | 33.92   | 36.13   | 40.23   | 39.26   | 40.33   |
> | R1-Distill-Qwen-14B  | Tokens   | 886.26  | 2289.21 | 3910.04 | 6035.36 | 6365.84 | 6167.72 | 5927.13 | 5463.18 | 5137.19 |
> |                      | Accuracy | 15.42   | 17.97   | 23.24   | 44.34   | 50.78   | 51.95   | 51.56   | 51.76   | 50.20   |
> | R1-Distill-Qwen-32B  | Tokens   | 266.07  | 296.68  | 1612.85 | 4696.14 | 5881.31 | 6391.68 | 6809.91 | 7128.37 | 7274.20 |
> |                      | Accuracy | 17.38   | 19.33   | 19.14   | 44.73   | 55.76   | 56.83   | 59.77   | 59.38   | 54.30   |
> | QwQ-32B              | Tokens   | 6490.33 | 6826.64 | 7368.09 | 7472.53 | 8087.62 | 8811.63 | 8763.01 | 9161.22 | 9862.67 |
> |                      | Accuracy | 48.43   | 58.79   | 59.38   | 59.76   | 60.55   | 60.96   | 60.16   | 59.38   | 55.86   |
>
> > **Questions**
>
> - **Chosen layers.** We apply the activation steering to every layer l following previous works [3], as explained in Appendix C.2. We will add this information in the main body of this paper.
>
> - **Choice of strength λ.** In most cases, we find λ = 0.05 or λ = 0.1 works fine on all the datasets. Usually, a default choice of λ = 0.05 can yield a good performance among all the benchmarks in most cases, and 0.1 sometimes works a little bit better. We will add the chosen strengths later in the table.
>
> - **Why fig2c has x=0 point but not the others?** Thanks for your detailed review. Sorry for making this confusing. We will unify this by adding x=0 for all the figures. The low performance when x=0 indicates that the same representation of end-of-reasoning token </think> at the input embedding layer can not yield any useful prediction (i.e., the place where we extract the representations), which is intuitive and reasonable.
>
> - **The average reasoning token number of overthink questions.** Yes. These overthink questions consistently yield more reasoning token numbers. We report the average reasoning token numbers in Table 4. The results indicate that these overthink attacks indeed make these LRMs spend more tokens for reasoning.
>
> **Table 4: Effect of overthink attack on the reasoning token number.**
>
> |                      | Original | + Overthink Attack |
> | -------------------- | -------- | ------------------ |
> | R1-Distill-Qwen-1.5B | 2456.88  | 4681.93            |
> | R1-Distill-Qwen-7B   | 2155.56  | 5386.25            |
> | R1-Distill-Qwen-14B  | 819.75   | 4539.21            |
> | R1-Distill-Qwen-32B  | 2041.38  | 7017.54            |
> | QwQ-32B              | 2819.03  | 12275.58           |
>
> [1] DAPO: An Open-Source LLM Reinforcement Learning System at Scale. Arxiv 2025.5.
>
> [2] SEAL: Steerable Reasoning Calibration of Large Language Models for Free. Arxiv 2025.4.
>
> [3] Refusal in Language Models Is Mediated by a Single Direction. NeurIPS 2024.

---

> > ### Comment · Reviewer_XAya · 2025-08-05
> >
> > Thanks for the detailed response. It made it clear that simple manipulation of the end-of-thought token is not sufficient for achieving similar performance. It also makes sense that the proposed method alters probability of thinking words like "Wait".
> >
> > But I am still not sure about the practicality of the method, but the new results on GPQA definitely makes it more convincing. For these reasons, I raise my score.

---

> > > ### Author Response · Authors · 2025-08-08
> > > **Thanks**
> > >
> > > We sincerely appreciate your time and valuable suggestions. We would include all your suggestions in our latest paper. Additionally, we would highlight our contribution in providing a potential mechanism support for practitioners for the first time.

---

### Official Review · Reviewer_qGsi · 2025-07-07

**Clarity:** 3
**Significance:** 2
**Originality:** 2
**Rating:** 3
**Confidence:** 3

**Summary:**

This paper investigates the internal mechanisms behind how Large Reasoning Models (LRMs) adjust their reasoning strength (i.e., the number of reasoning tokens generated) based on task difficulty. The authors propose that LRMs encode a pre-allocated directional vector in the activation space before generating any reasoning tokens. This vector’s magnitude correlates with the difficulty of the task and modulates reasoning length. Through linear probing and activation steering, the paper shows: 1) Reasoning length is predictable from pre-generation activations. 2) A consistent direction in activation space correlates with task difficulty. 3) Manipulating this vector causally affects reasoning length and performance. 4) The vector modifies the logits of the end-of-reasoning token </think>. Two applications are demonstrated: predicting overthinking before generation and reducing unnecessary reasoning on simple tasks for efficiency.

**Questions:**

see the Weaknesses part

**Ethical Concerns:**

["NO or VERY MINOR ethics concerns only"]

**Final Justification:**

I remain my score as the practical applicability of the approach remains unclear given the current results. The mean absolute error of the predicted token count is very high, and the reduction in error after adding 500 additional training samples on MATH is relatively small. Moreover, enabling overthinking detection using this approach requires practitioners to manually set token-length thresholds for each question or train a difficulty classifier.

**Limitations:**

yes

**Quality:**

3

**Strengths And Weaknesses:**

## Strength

- Addresses an important mechanistic question about reasoning strength in LRMs with practical applications for overthinking detection and efficient inference
- Clear research questions and logical progression from hypothesis to validation.
- Well-designed experimental framework using linear probing to test the pre-planning hypothesis.
- The authors provide validation across multiple model sizes and datasets. The results are promising, e.g., moderating positive steering shows potential in enhancing performance across other math datasets, including MATH500, AIME, and OlympiadBench.


## Weaknesses


1. Scope & Generalizability:
- Experiments focus primarily on mathematical reasoning (MATH). Unclear if findings generalize to other reasoning domains (logical reasoning, coding, etc.). Given the current results, authors might consider revising the scope to acknowledge this as a focused study on mathematical reasoning, rather than general claims about reasoning mechanisms in LRMs.
- Heavy reliance on Qwen-based models with limited validation on other model families (e.g., LLaMA, Mistral, GPT-based models)

2. Methodological/Experimental Design Justification:
- For linear probing, the choice of residual stream activations at each layer lacks justification. Why not use input embeddings at each layer or other activation representations?
- Equation 1 presents a regularized loss with hyperparameter $\alpha$  without discussion of how this hyperparameter $\alpha$ is selected. Additionally, the rationale for choosing Lasso (L1) regression (e.g., over L2 regularization) is not provided.
- For activation steering (Section 4.2), it's unclear whether steering applies to every transformer layer $l$ or only specific layers
- Table 1 doesn't specify how the steering strength $\lambda$ is selected across different benchmarks. Are values consistent across tasks or require task-specific tuning?

3. Analysis:
- Section 3.2: The linear predictor is both trained and tested on MATH. It's unclear whether separate predictors are trained for each difficulty level or if a single joint predictor handles all difficulties. Can this predictor accurately predict token lengths for other mathematical reasoning tasks beyond MATH?

- Figure 2 reports only correlation coefficients without accuracy metrics. For practical understanding, concrete examples would be helpful: given an actual token length y for question x, what is the predicted length y' and the absolute error?

- While Section 3.2 observes that deeper layers yield better predictions, but it lacks explanation for why reasoning planning capabilities develop in later layers.

- Figure 7 appears inconsistent with Figures 2/4: while later layers show better prediction accuracy in Figure 2, Figure 7 shows seemingly random results across layers without clear layer-specific accuracy patterns.

4. Baseline and Comparison Issues:
- Figure 3 and lines 181-182 claim high cosine similarity between vectors, but without proper baselines, these absolute similarity values lack context. A baseline comparing difference-in-mean vectors from randomly sampled question pairs might strengthen the argument.

- Figures 6 and 7 should be presented together for direct comparison between actual and predicted token lengths. The predicted token lengths in Figure 7 (≈4000, 3500, 2000) don't align well with actual lengths from the models (≈5000, 4000, 8000 under λ=0.2), suggesting prediction limitations that need addressing.


5. Application Practicality
-  Section 5.1's overthinking detection is interesting, but how should practitioners determine token length thresholds for detecting attacks? The current approach compares predictions before/after attacks, but real-world scenarios require determining if a single question represents vanilla or attack conditions without prior knowledge.




6. Related work:
- The probing analysis would benefit from discussing related work on reasoning interpretability in language models, such as [1]

7. Minor:
- Typo "b" in line 107
- Inconsistent difficulty level numbering: line 176 mentions difficulty 0, while Equation 2 and line 178 refer to easiest difficulty level as 1.



Reference

- [1] Physics of Language Models: Part 2.1, Grade-School Math and the Hidden Reasoning Process. ICLR 2025

---

> ### Author Rebuttal · Authors · 2025-07-31
>
> We sincerely appreciate your detailed evaluation and valuable suggestions! We respond to your questions as follows:
>
> ---
>
> > **Scope & Generalizability**
>
> - **Other reasoning domains.** We discussed the generalization capabilities of our findings on other general domains beyond MATH, such as AlpacaEval and MMLU in Section 5, where our findings of reasoning strength prediction and strength control also hold on these two datasets. We also add one more experiment on the complex logic reasoning task GPQA diamond to better demonstrate the generalization capabilities. As shown in Table 1, the number of reasoning tokens generally exhibits a similar pattern with the λ, indicating the generalization of this pre-allocation vector to other domains. Additionally, proper positive steering can also bring performance improvements.
>
> **Table 1: The effect of steering and performance on GPQA diamond**
>
> |λ||-0.2|-0.15|-0.1|-0.05|0|0.05|0.1|0.15|0.2|
> |-|-|-|-|-|-|-|-|-|-|-|
> |R1-Distill-Qwen-1.5B|Tokens|959.44|2930.29|5659.00|6677.72|7428.22| 7690.48|7902.20|7805.64|7784.97|
> ||Accuracy|8.01|6.05|8.20|12.30|13.86|16.80|17.58|16.80|19.53|
> |R1-Distill-Qwen-7B|Tokens|1022.10|2755.88|5250.92|6243.68|6998.01|7326.04|7875.52|8103.89|8380.32|
> ||Accuracy|5.66|12.30|22.27|28.13|33.92|36.13|40.23|39.26|40.33|
> |R1-Distill-Qwen-14B|Tokens|886.26|2289.21|3910.04|6035.36|6365.84| 6167.72|5927.13|5463.18|5137.19|
> ||Accuracy|15.42|17.97|23.24|44.34|50.78|51.95|51.56|51.76|50.20|
> |R1-Distill-Qwen-32B|Tokens|266.07|296.68|1612.85|4696.14|5881.31 | 6391.68|6809.91|7128.37|7274.20|
> ||Accuracy|17.38|19.33|19.14|44.73|55.76|56.83|59.77|59.38|54.30|
> |QwQ-32B|Tokens|6490.33|6826.64|7368.09|7472.53|8087.62|8811.63|8763.01|9161.22|9862.67|
> ||Accuracy|48.43|58.79|59.38|59.76|60.55|60.96|60.16|59.38|55.86|
>
> - **Other model families.** We add experiments on another kind of LRM DeepSeek-R1-Distill-Llama-8B. All of our findings also hold on this model, and we will add more results in our paper. We illustrate some datapoints of our main observations on DeepSeek-R1-Distill-Llama-8B. Interestingly, Llama is more sensitive to the steering strength λ.
>
> **Table 2: Prediction on middle layers**
>
> |Layer|25|26|27|28|29|30|
> |-|-|-|-|-|-|-|
> | R|0.8481|0.8467|0.8442|0.8464|0.8423|0.8430|
>
> **Table 3: Cosine similarity between pre-allocated vectors on layer 31**
>
> ||r2<-1|r3<-1|r4<-1|r5<-1|
> |-|-|-|-|-|
> |**r2<-1**|1.00|0.98|0.98|0.90|
> |**r3<-1**|0.98|1.00|1.00|0.93|
> |**r4<-1**|0.98|1.00|1.00|0.94|
> |**r5<-1**|0.90|0.93|0.94|1.00|
>
>
> **Table 4: Mean cosine similarities on middle layers**
>
> |Layer|25|26|27|28|29|30|
> |-|-|-|-|-|-|-|
> | R|0.9681|0.9661|0.8873|0.9620|0.9660|0.9124|
>
> **Table 5: Activation Steering**
>
> |λ||-0.02|-0.015|-0.01|-0.005|0|0.005|0.01|0.015|0.02|
> |-|-|-|-|-|-|-|-|-|-|-|
> |MATH500|Tokens|3042.42|3391.56|3312.67|3478.20|3457.01|3484.90|3530.19  |3702.91|3654.19|
> ||Accuracy|75.18|80.00|81.78|83.55|82.80|83.38|83.98|84.88|84.48|
> |AIME2024|Tokens|10366.34|10910.79|10458.87|10750.82| 11049.53 | 10774.68 |10888.95|11553.91|11622.77|
> ||Accuracy|26.67|32.08|34.78|38.75|38.96|38.75|37.08|42.50|41.67|
> |OlympiadBench|Tokens|6926.34|7174.21|7254.21|7265.71|7234.64|7338.05| 7522.05|7615.02|7345.39|
> ||Accuracy|43.69|48.24|49.94|50.13|49.76| 49.59|50.41|51.20|49.30|
>
> > **Methodological / Experimental Design Justification:**
>
> - **Choice of residual stream activations.** Sorry for this confusion. Actually, we use the input embeddings at each layer as you have mentioned. These input embeddings are called the (input) residual stream in previous literature [1, 2], where representation at this place is a common usage. We will make this clearer by adding references and add explanation for this in our paper.
>
> - **Selection of α and rationale for Lasso regression.** Actually, we do not carefully tune this hyperparameter α since tuning this hyperparameter at each layer is laborious. Therefore, we just choose a moderate α=10 that works fine on DeepSeek-R1-Distill-Qwen-7B and adopt this α for all the models. It is possible that we can further improve the prediction accuracy by carefully tuning α. For the choice of L1 regression, we found that both L1 and L2 work similarly. Therefore, we choose L1 following the choice of previous work [3]. We show the comparison experiments on the 50th layer of DeepSeek-R1-Distill-Qwen-32B in Table 6 and Table 7. When α is small, the regression starts to overfit, showing a high value of R on the training set and a low value on the test set.
>
> **Table 6: Comparison of α**
>
> |α|0.1|1|10|100|
> |-|-|-|-|-|
> |R (Train)|0.9060|0.9025|0.8795|0.8491|
> |R (Test)|0.6034|0.7424|0.8413|0.8341|
>
> **Table 7: Comparison with L1 and L2 regularization.**
>
> ||L1|L2|
> |-|-|-|
> |R (Test)|0.8413|0.8356|
>
> - **Layers of activation steering.** We apply the activation steering to every layer l following previous works [2], as explained in Appendix C.2. We will add this information in the main body of this paper.
>
> - **Choice of strength λ.** In most cases, we find λ = 0.05 or λ = 0.1 works fine on all the datasets. Usually, a default choice of λ = 0.05 can yield a good performance among all the benchmarks in most cases, and 0.1 sometimes works a little bit better. We will add more details in our paper.
>
> > **Analysis**
>
> - **Training of linear probing.** We train a single predictor handling all the difficulties. We also provide experiments predicting other mathematical reasoning tasks in Table 8. Results show a still high R value, indicating the generalization capabilities.
>
> **Table 8: Prediction on other benchmarks**
>
> ||MATH500|OlympiadBench|
> |-|-|-|
> |R (Test)|0.8234|0.7823|
>
> - **Accuracy metrics.** We report the mean absolute error (MAE) as the accuracy metric. The predictor in Figure 1 yields MAE = 591.37 for y under 4000 tokens. Meanwhile, we add one more experiment to demonstrate that this prediction is non-trivial. We randomly permute the label y (i.e., the actual reasoning token number) and make a prediction again. The MAE under this random prediction setting is 1428.72, which is significantly higher than our prediction.
>
>   We would like to highlight again that this work reveals whether and how LRMs plan the reasoning strength rather than making this prediction as accurate as possible.  It is worth exploring in the future for improving the accuracy by adding more training data, adopting a more complicated prediction model structure, and carefully tuning the hyperparameters. For example, if we merely add 500 more training datapoints from MATH500, the MAE further reduces from 591.37 to 563.42.
>
> - **Reasoning capabilities in later layers.** Previous mechanism explanation works suggest that the early layers of language models may be responsible for low-level tasks such as syntax and grammar, and more complicated understanding capabilities are developed in the later layers [4]. Recent findings also suggest that reasoning capabilities may be developed in the later layers [5], which provides more support for our findings.
>
> - **Figure 7 appears inconsistent with Figures 2/4 & Figures 6 and 7 should be presented together.** Thanks for the detailed review, and also sorry for the confusion. Actually, Figure 7 depicts the effect of the steering vector with strength λ=0.2, rather than the prediction for the whole response. We apologize for using a misleading annotation y_hat in Equation 4, and we will revise this in our latest paper. For Figure 7, we suggest mostly focusing on the average predictions ≈ 1700, 1500, and 500, which are similar to the changes in the reasoning token numbers from a zero strength to a strength λ=0. Moreover, we attribute the disturbance across layers to the fact that we may find the main direction vector for strength planning, while there may exist sub-directions for more fine-grained control, as has been evidenced in similar works [2, 6].
>
> > **Baseline and Comparison Issues**
>
> - **Cosine similarity**. We add experiments comparing with difference-in-mean vectors from randomly sampled question pairs. We randomly sample three sub-dataset, each with 2000 datapoints, and calculate the mean activation of them, denoted as diff_1, diff_2, and diff_3. Then we calculate the cosine similarity between diff_3 - diff_1 and diff_2 - diff_1 as the comparison. This random cosine similarity is generally around 0.45, with 0.4634 at the same layer of Figure 3.a. This result further supports the existence of the pre-allocation vector for reasoning strength planning.
>
> > **Application Practicality**
>
> Thanks for your interest. As we have highlighted in the first paragraph in Section 5, to make this approach in practice, more efforts are required. One possible solution for making this overthink detection in practice is probably that we can additionally train a difficulty classifier. Once this problem is identified as simple, but the reasoning strength prediction yields an overlong prediction, we can provide a warning for the possible occurrence of overthink. This is worth exploration in the future, and we think it deserves the effort for another research paper.
>
> > **Related Works**
>
> Thanks for this suggestion. We believe the idea in this paper that LLMs know their correctness can further provide support for our claims that LRMs know their reasoning strengths in advance. We will cite this paper and discuss it in our latest paper.
>
> > **Minor**
>
> Thanks for your detailed suggestions. We will revise them in the latest paper.
>
>
> [1] A Mathematical Framework for Transformer Circuits. Anthropic 2021.
>
> [2] Refusal in Language Models Is Mediated by a Single Direction. NeurIPS 2024.
>
> [3] Language models represent space and time. ICLR 2024.
>
> [4] Safety Layers in Aligned Large Language Models: The Key to LLM Security. ICLR 2025.
>
> [5] Detection and Mitigation of Hallucination in Large Reasoning Models: A Mechanistic Perspective. Arxiv 2025.5.
>
> [6] The Hidden Dimensions of LLM Alignment: A Multi-Dimensional Safety Analysis. ICML 2025.

---

> > ### Comment · Reviewer_qGsi · 2025-08-05
> >
> > Thank you for the response.
> >
> > The detailed explanation resolves most of my concerns, and I appreciate the additional experiments on more datasets and models, as well as the justification for hyperparameter selection.
> >
> > That said, I still have some concerns about the practical applicability of the approach given the current results. The mean absolute error of the predicted token count remains high, and the reduction in error after adding 500 additional training samples on MATH is relatively small. Moreover, as the authors note, enabling overthinking detection requires practitioners to either manually set token-length thresholds or train a difficulty classifier, which limits its application in practice.

---

> > > ### Author Response · Authors · 2025-08-05
> > > **Thanks and Follow-Up Responses**
> > >
> > > Thanks for your detailed review response.
> > >
> > > As we have explained in our paper and rebuttal, the main contribution of this paper is not proposing a method for perfect prediction. Instead, we aim to reveal the underlying mechanism of reasoning strength planning in LRMs, which would benefit the LRM research.
> > >
> > > The prediction precision can be improved by adopting more advanced models or adding more data. The improvement of adding 500 training examples is non-trivial, given the amount of 500 examples are only a small part of our total training data (i.e., 6750). Additionally, we would like to highlight the MAE is compared with the fact that these reasoning token number often exceeds thousands, indicating the predictability.
> > >
> > > Regarding overthink detection, we highlight our contribution in providing potential mechanism support for practitioners, putting forward this possibility for the first time. Meanwhile, papers where such mechanism explanations provide support for subsequent methods are common [1, 2].
> > >
> > > We hope our rebuttal efforts and improvements can be considered. If our responses address most questions and our refinements are acknowledged, we would appreciate an adjusted evaluation.
> > >
> > >
> > > [1] Refusal in Language Models Is Mediated by a Single Direction. NeurIPS 2024.
> > > [2] Programming Refusal with Conditional Activation Steering. ICLR 2025.

---

### Comment · Area_Chair_YCXV · 2025-08-03
**Please discuss the paper asap**

Dear reviewers,

Now the rebuttal is available. Thanks reviewer ZwPk for starting the discussion. Others, please discuss with authors and among reviewers asap.

Please try to come to a consensus on the key issues even though the rating can be different. Please feel free to let me know how I can help.

Best,

Your AC

---

### Note · Authors · 2025-08-12

Dear Reviewers and ACs,

We sincerely appreciate your efforts and time during the rebuttal. Your suggestions have greatly helped us refine our work.

In this work, we systematically analyze the reasoning strength planning mechanism in large reasoning models. We reveal that LRMs plan their reasoning strength in advance, even before the generation of the first token, using the magnitude of a pre-allocation vector for controlling the reasoning strength. We are delighted that reviewers recognized the **importance of the research topic** (Reviewers $\color{red}{\text{qGsi}}$, $\color{blue}{\text{XAya}}$, and $\color{green}{\text{1xAp}}$), the **comprehensive analysis** (Reviewers $\color{red}{\text{qGsi}}$, $\color{green}{\text{1xAp}}$, and $\color{orange}{\text{ZwPk}}$), and the **well-organized paper writing** (Reviewer $\color{orange}{\text{ZwPk}}$). The reviewers also raised insightful questions, which we have responded to as follows:

- **Generalization on Other Domains:** We have added two more experiments on two different reasoning tasks (i.e., GPQA diamond and LiveCodeBench) beyond math to demonstrate the generalization capabilities (Reviewers $\color{red}{\text{qGsi}}$, $\color{blue}{\text{XAya}}$, $\color{green}{\text{1xAp}}$, and $\color{orange}{\text{ZwPk}}$).
- **Generalization on Other LRMs:** We have conducted additional experiments on another LRM, DeepSeek-R1-Distill-Llama-8B, demonstrating that our findings can be generalized to different LRM backbones (Reviewer $\color{red}{\text{qGsi}}$).
- **More Clarification:** We have added more discussion about prediction accuracy (Reviewers  $\color{red}{\text{qGsi}}$ and $\color{blue}{\text{XAya}}$), practical application (Reviewers  $\color{red}{\text{qGsi}}$ and $\color{blue}{\text{XAya}}$), the more comprehensive mechanism of logits control (Reviewer $\color{blue}{\text{XAya}}$), more explanation for figures (Reviewers  $\color{red}{\text{qGsi}}$, $\color{blue}{\text{XAya}}$, $\color{green}{\text{1xAp}}$, and $\color{orange}{\text{ZwPk}}$), and others.
- **More Details about the Implementation:** We have added more discussion about the implementation details, including the layers for steering (Reviewers  $\color{red}{\text{qGsi}}$ and $\color{blue}{\text{XAya}}$), the selection of steering strength (Reviewers  $\color{red}{\text{qGsi}}$, $\color{blue}{\text{XAya}}$, and $\color{orange}{\text{ZwPk}}$), and others.

Thanks again for your invaluable suggestions and support.

Best,

The Authors

---

### Decision · Program_Chairs · 2025-09-17

**Decision:**

Accept (poster)

**Comment:**

**Recommendation:** Need Discussion

### (a) Summary of the Paper

This paper investigates the internal mechanism by which Large Reasoning Models (LRMs) allocate "reasoning strength," defined as the number of reasoning tokens they generate. The central and novel finding is that this allocation is pre-planned: the model encodes the expected reasoning length in a specific directional vector within its activations *before* token generation begins. The authors demonstrate this through linear probing to predict reasoning length from pre-generation activations and use activation steering to causally manipulate it, showing that increasing the vector's magnitude leads to longer reasoning chains and vice-versa. The paper concludes by demonstrating two potential applications: detecting "overthinking" behavior and improving inference efficiency on simple tasks.

### (b) The Case for Acceptance (The Majority View)

The paper received strong support from two reviewers (1xAp, ZwPk), with a third leaning toward acceptance (XAya). The primary argument for acceptance is that the paper provides a **novel and significant mechanistic explanation** for a widely observed but poorly understood phenomenon in LRMs. This type of interpretability work, which moves beyond correlation to causal intervention, is a valuable contribution to the field. The experimental methodology, combining linear probing with activation steering, is considered solid and well-executed.

Furthermore, the authors were highly responsive during the rebuttal, conducting substantial new experiments that successfully addressed the most critical initial weakness: **generalizability**. By adding results on a different model family (DeepSeek-Llama) and entirely different reasoning domains (GPQA for logic/science, LiveCodeBench for code), they demonstrated that their core findings are not mere artifacts of the Qwen models or mathematical reasoning tasks. This effort was highly convincing to the supporting reviewers, solidifying their positive stance. They argue that the paper's insights open up promising new avenues for controlling and understanding LRMs, even if the immediate practical applications are not yet fully mature.

### (c) The Case for Rejection (The Dissenting View)

Despite the positive feedback, two reviewers (qGsi, XAya) maintain significant reservations, primarily centered on the **limited practical significance and applicability of the findings**. While acknowledging the novelty of the mechanistic discovery, they argue that the proposed applications are not yet compelling enough to warrant publication at NeurIPS.

The key points against acceptance are:

1.  **Limited Practical Utility of Steering:** The performance gains from steering the "reasoning strength" vector are marginal and fragile. As shown in the paper's own results, while a small positive steering can slightly improve performance, too much steering leads to a performance drop. There is no principled way presented to determine the optimal steering strength (`λ`), making it difficult to use in practice.
2.  **Unconvincing Applications:** The two demonstrated applications are seen as underdeveloped. The "overthinking detection" relies on predicting the reasoning length, but the linear probe's predictions have a high Mean Absolute Error, limiting their reliability. Furthermore, deploying this in a real-world scenario would require an external component, like a difficulty classifier, which is outside the scope of the paper. The "efficient reasoning" application is similarly constrained by the lack of a method to dynamically choose the correct negative steering strength.
3.  **The Mechanistic Finding Alone May Not Be Sufficient:** The core of this argument is that while the paper makes an interesting scientific observation, it doesn't translate this observation into a robust, usable technique.

### (d) Key Points for AC Discussion

The review process has surfaced a clear and fundamental disagreement about what constitutes a sufficient contribution for NeurIPS in the context of mechanistic interpretability. The discussion points are:

1.  **Significance of Mechanistic Insight vs. Practical Utility:** This is the central point of contention. Is a novel, well-supported mechanistic explanation of an important phenomenon (how LRMs plan reasoning length) a sufficient contribution on its own? Or must it be accompanied by a clear, robust, and practical application to be considered high-impact?

2.  **Evaluating the Practical Applications:** How should we assess the proposed applications? The supporters view them as valuable proofs-of-concept that open new research directions. The skeptics view them as premature and not practically viable in their current form (due to high prediction error, lack of a principled method for setting `λ`, and the need for external components).

3.  **The Strength of the Causal Claim:** The activation steering experiments provide causal evidence. How strong is this evidence, and how much does it contribute to the paper's significance? Does the fact that steering can also *harm* performance weaken the claim of having found a clean, controllable "reasoning strength" vector, or does it simply highlight the complexity of the internal mechanism?